# Two-loop splitting in double parton distributions

**Markus Diehl[1⋆], Jonathan R. Gaunt[2†], Peter Plößl[3‡] and Andreas Schäfer[3∘]**

**1** Deutsches Elektronen-Synchrotron DESY, 22603 Hamburg, Germany
**2** CERN Theory Division, 1211 Geneva 23, Switzerland
**3** Institut für Theoretische Physik, Universität Regensburg, 93040 Regensburg, Germany

⋆ markus.diehl@desy.de, † jonathan.richard.gaunt@cern.ch,
‡ peter.ploessl@physik.uni-regensburg.de, ∘ andreas.schaefer@physik.uni-regensburg.de

## Abstract

Double parton distributions (DPDs) receive a short-distance contribution from a single parton splitting to yield the two observed partons. We investigate this mechanism at next-to-leading order (NLO) in perturbation theory. Technically, we compute the two-loop matching of both the position and momentum space DPDs onto ordinary PDFs. This also yields the $1 \to 2$ splitting functions appearing in the evolution of momentum-space DPDs at NLO. We give results for the unpolarised, colour-singlet DPDs in all partonic channels. These quantities are required for calculations of double parton scattering at full NLO. We discuss various kinematic limits of our results, and we verify that the $1 \to 2$ splitting functions are consistent with the number and momentum sum rules for DPDs.

Copyright M. Diehl *et al*.
This work is licensed under the Creative Commons
Attribution 4.0 International License.
Published by the SciPost Foundation.

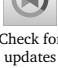

# 1 Introduction

Consider the integrated cross section for the production of some final state with associated hard scale $Q$ at a hadron collider. As is well known, the dominant mechanism for this is single parton scattering (SPS), where one parton from each proton collides to form the state of interest. If the final state can be divided into two subsets $A$ and $B$ with associated hard scales $Q_A$ and $Q_B$, then another mechanism is two separate parton-parton collisions, one yielding $A$ and the other yielding $B$. This is called double parton scattering, or DPS. The integrated cross section for this mechanism is suppressed compared to that for SPS by $\Lambda^2/Q^2$, where $\Lambda$ denotes a hadronic scale and $Q = \min(Q_A, Q_B)$. Despite this, there are several motivations to study this process, and DPS occupies a special place amongst the power suppressed corrections for the production of $A + B$. First, DPS may compete with SPS when the latter is suppressed by small or multiple coupling constants, with a well-known example being same-sign $W$ pair production [1–6]. When followed by leptonic $W$ decays, this is an important process, since same-sign dilepton production is a search channel for new physics (see e.g. [7–9]). Second, although DPS is a power correction at the level of the integrated cross section, in differential cross sections there can be regions in which DPS is competitive with SPS. In particular, for small transverse momenta of $A$ and $B$, DPS and SPS contribute at the same power [10–13]. For several processes one also finds that DPS is an important contribution at large rapidity separation of $A$ and $B$ (see e.g. [7, 14, 15]). Third, the DPS cross section scales roughly as the fourth power of a parton distribution, whilst SPS scales as the second power (and other $\Lambda^2/Q^2$ corrections scale as the third power). Thus for a given final state, as we increase the collider energy and probe smaller parton momentum fractions $x$, DPS grows faster than the other contributions, and hence is especially relevant at the LHC and future, higher energy colliders. Finally, DPS is an interesting process to study in its own right: it provides information on the correlation between partons in the proton and thus reveals an aspect of hadron structure that remains inaccessible in single parton distributions.

A variety of DPS processes have been investigated at the ISR [16], the CERN SPS [17], the Tevatron [18–26], and the LHC [4, 27–37] (see e.g. figure 4 of [35] and figure 15 of [38] for overviews). These measurements cover final states with jets, electroweak gauge bosons, heavy

quarkonia, and open charm. The DPS contribution to pair production of heavy gauge or Higgs bosons (such as $W^+W^-$ or $HZ$) is estimated to be small, but may well become important in the high-luminosity phase of the LHC, given the overarching goal to probe electroweak symmetry breaking with highest possible precision. A detailed review of experimental, phenomenological, and theoretical aspects of DPS can be found in [39].

Either one or both parton pairs in a DPS process can arise from a single parton via a perturbative "1 → 2" branching process. At next-to-leading order (NLO) and above, this branching may produce additional partons that are emitted into the final state, in addition to the two "active" partons that subsequently enter hard collisions. The process in which we have 1 → 2 splittings in both protons overlaps with a loop correction to SPS, in the sense that a given Feynman graph corresponds to either DPS or SPS depending on the momenta of its internal lines. Likewise, the process where we have a 1 → 2 splitting in just one proton overlaps with a "twist 2 × twist 4" power-suppressed process. Developing a theoretical QCD framework for DPS that includes 1 → 2 splitting but avoids double counting between SPS and DPS (and the twist 2 × twist 4 mechanism) is quite nontrivial. First approaches to this problem are given in [13, 40–42]. More recently, some of us developed a new formalism [43] that possesses several advantages over the previous approaches. This involves a description of the DPS cross section in terms of so-called position space double parton distributions (DPDs) $F_{a_1 a_2}(x_1, x_2, \boldsymbol{y})$, where $\boldsymbol{y}$ is the separation in transverse space between the two partons.

One of the advantages of the approach in [43] is that it can be formulated at all orders in $\alpha_s$, with higher-order contributions that can be practicably computed. This opens the perspective to performing DPS calculations at full next-to-leading order accuracy. The framework of [43] is designed to make maximal re-use of quantities that are already computed for SPS. The partonic scattering cross sections $\hat{\sigma}$ for DPS and for SPS are identical. NLO effects in parton emissions from the incoming legs are correctly described by including the known NLO corrections to the DGLAP splitting kernels [44–52] in the corresponding kernels of the "double DGLAP" equation, which describes the scale evolution of $F_{a_1 a_2}(x_1, x_2, \boldsymbol{y})$. A missing key ingredient, specific to DPS, is the effect of the 1 → 2 splitting at NLO. This is the subject of the present paper. What is technically required in the notation of [43] is the two-loop kernel $V_{a_1 a_2, a_0}(x_1, x_2, \boldsymbol{y})$ for the perturbative matching of the DPDs onto the PDFs $f_{a_0}(x)$ at small $\boldsymbol{y}$.

Why should one consider performing such calculations? Apart from unknown higher-order corrections, a major source of uncertainty in DPS cross section calculations is the lack of knowledge of the nonperturbative "input" DPDs. Unlike ordinary parton distributions (PDFs), these are very poorly constrained due to the limited experimental information on DPS. One can expect knowledge of the DPDs to improve in future, as further information emerges from experiment, and as this information is combined with theoretical constraints [53, 54] and information emerging from lattice computations (see [55] for work in this direction). However, it is unlikely that the situation for DPDs will approach that for PDFs in the near-term future. Despite a considerable uncertainty on the shape of the nonperturbative DPDs in general, there are a number of motivations for analysing DPS at NLO accuracy. Concerning the partonic cross sections $\hat{\sigma}$, it is well known from SPS that in many cases their LO expressions give only a rough approximation, especially for final states containing jets. On the side of DPDs, the 1 → 2 splitting mechanism requires only PDFs as a nonperturbative input and can thus be brought under good theoretical control. Several studies have emphasised the quantitative impact of this mechanism on DPS [43, 56–62]. It is thus natural to go beyond the LO approximation for this contribution and compute the splitting at order $\alpha_s^2$. Moreover, for some parton combinations, this is the first order that gives a nonzero result. An example is the $u\bar{d}$ distribution, which is in particular relevant for same-sign $W$ pair production. The recent study in [63] finds that the splitting mechanism has some impact on observables even in such a situation. A general aim of going beyond LO is to get a handle on the perturbative convergence. As previously

mentioned, favourable regions for DPS typically involve small $x$ values, and in this region there is potential that large logarithms $\log(1/x)$ can have a significant impact, and even spoil the perturbative convergence. By studying the effect of the NLO corrections on DPDs, and more generally on DPS luminosities and cross sections, we can get an indication of whether there are convergence issues caused (for example) by small $x$ logarithms. Finally, it is quite possible that by constructing appropriate ratios of cross sections (involving, for example, the same parton flavours but different scales), one may minimise the dependence on the nonperturbative input and more directly test the perturbative part of the cross section. In this case, NLO computations clearly should improve the overall accuracy.

In this paper we compute the matching coefficients $V_{a_1 a_2, a_0}(x_1, x_2, \boldsymbol{y})$ at two loops for all partonic channels (i.e. for all combinations of parton labels $a_0, a_1, a_2$). We restrict ourselves to the matching coefficients for the colour-singlet, unpolarised DPDs. These are expected to be the dominant DPDs at small $x$ and high $Q$ [64, 65]; in fact they are the only DPDs considered in most phenomenological analyses of DPS. The coefficients for the colour interference and polarised DPDs will be presented in future work.

Aside from DPDs $F_{a_1 a_2}(x_1, x_2, \boldsymbol{y})$ in position space, one can also consider momentum space DPDs $F_{a_1 a_2}(x_1, x_2, \boldsymbol{\Delta})$. The variable $\boldsymbol{\Delta}$ is Fourier conjugate to $\boldsymbol{y}$ and represents the difference in transverse momentum between a parton entering the DPD operator in the amplitude and the corresponding parton in the conjugate amplitude. A description of DPS in terms of momentum space DPDs is considered in a number of works [12, 13, 56, 61, 66]. Furthermore, the momentum-space DPDs at $\boldsymbol{\Delta} = \boldsymbol{0}$ are the objects that satisfy the number and momentum sum rules of [53], as was discussed in [54]. Similarly to the position space DPDs at small $\boldsymbol{y}$, momentum space DPDs at large $\boldsymbol{\Delta}$ can be matched onto PDFs with perturbative kernels $W_{a_1 a_2, a_0}(x_1, x_2, \boldsymbol{\Delta})$. For the momentum space DPDs, the effect of $1 \to 2$ splittings also shows up in the evolution equations, which contain an inhomogeneous term where the PDFs are convolved with "$1 \to 2$ evolution kernels" $P_{a_1 a_2, a_0}(x_1, x_2)$ [11, 66–68]. In [54] it was shown that the kernels $P_{a_1 a_2, a_0}$ are related by sum rules to the usual DGLAP splitting functions at all orders in $\alpha_s$. Finally, one can introduce matching kernels $U_{a_1 a_2, a_0}(x_1, x_2)$ that allow one to compute the distributions $F_{a_1 a_2}(x_1, x_2, \boldsymbol{\Delta} = \boldsymbol{0})$ that correspond to a given set of position space DPDs. This allows one to make use of the sum rule constraints of [53] also if one calculates DPS cross sections in terms of position space DPDs. The computation of the kernels $U_{a_1 a_2, a_0}$ at the leading order level is given in section 7 of [43] and will be extended to NLO in the present work.

This paper is organised as follows. In section 2 we define the kernels $V_{a_1 a_2, a_0}$, $W_{a_1 a_2, a_0}$ and $P_{a_1 a_2, a_0}$, analyse their renormalisation scale behaviour, and derive relations between them. The general structure of the matching between momentum and position space DPDs is discussed in section 3. In section 4 we give details of the two-loop computation of the matching kernels. We present our results and discuss various properties of them in section 5. A brief summary of this paper is given in section 6.

# 2 General analysis: ultraviolet behaviour and scale dependence

In the next two subsections we briefly review the definitions of bare and of renormalised DPDs, setting up our notation on the way. In section 2.3 we then derive how renormalisation affects the splitting kernels for DPDs and their logarithmic dependence on the renormalisation scale $\mu$. This will later allow us to obtain renormalised kernels from the computation of bare Feynman graphs for the relevant splitting processes.

## 2.1 Basic definitions and notation

To begin with let us recall that bare, i.e. unrenormalised, DPDs for partons $a_1$ and $a_2$ are defined by matrix elements

$$
F_{B,a_1a_2}(x_1, x_2, \boldsymbol{y}) = (x_1 p^+)^{-n_1} (x_2 p^+)^{-n_2} 2p^+ \int dy^- \frac{dz_1^-}{2\pi} \frac{dz_2^-}{2\pi} e^{i(x_1 z_1^- + x_2 z_2^-)p^+}
$$
$$
\times \langle p | \mathcal{O}_{a_1}(y, z_1) \mathcal{O}_{a_2}(0, z_2) | p \rangle \big|_{z_1^+ = z_2^+ = y^+ = 0, \boldsymbol{z}_1 = \boldsymbol{z}_2 = \boldsymbol{0}}, \tag{1}
$$

where $n_i = 0$ for quarks and antiquarks and $n_i = 1$ for gluons. We use light-cone coordinates $v^\pm = (v^0 \pm v^3)/\sqrt{2}$ for any four-vector $v^\mu$ and write its transverse part in boldface, $\boldsymbol{v} = (v^1, v^2)$. It is understood that the transverse proton momentum $\boldsymbol{p}$ is zero and that the proton polarisation is averaged over. The twist-two operators

$$
\mathcal{O}_q(y, z) = \frac{1}{2} \bar{q}(y - z/2) \gamma^+ q(y + z/2), \qquad \mathcal{O}_{\bar{q}}(y, z) = -\frac{1}{2} \bar{q}(y + z/2) \gamma^+ q(y - z/2),
$$
$$
\mathcal{O}_g(y, z) = G^{+i}(y - z/2) G^{+i}(y + z/2) \tag{2}
$$

for unpolarised partons are constructed from bare field operators and are familiar from the definition of bare PDFs,

$$
f_{B,a}(x) = (x p^+)^{-n} \int \frac{dz^-}{2\pi} e^{ixz^- p^+} \langle p | \mathcal{O}_a(0, z) | p \rangle \big|_{z^+ = 0, \boldsymbol{z} = \boldsymbol{0}}. \tag{3}
$$

In the present work, we consider colour-singlet DPDs, so that the colour indices of the quark or gluon fields in (2) are implicit and to be summed over. The form (2) holds in light-cone gauge $A^+ = 0$, and in other gauges Wilson lines need to be inserted between the two parton fields. The DPDs in (1) depend on the transverse distance $\boldsymbol{y}$ between the two partons. Distributions in transverse momentum representation are obtained by a Fourier transform

$$
F_{B,a_1a_2}(x_1, x_2, \boldsymbol{\Delta}) = \int d^{2-2\epsilon}\boldsymbol{y}\, e^{i\boldsymbol{y}\boldsymbol{\Delta}} F_{B,a_1a_2}(x_1, x_2, \boldsymbol{y}), \tag{4}
$$

where it is understood that bare quantities are evaluated in $d = 4 - 2\epsilon$ space-time dimensions. After ultraviolet renormalisation, which will be discussed in section 2.2, one obtains finite parton distributions and can set $d = 4$. Due to their different ultraviolet behaviour, renormalised DPDs in position and in momentum space are *not* Fourier conjugate to each other, unlike their bare counterparts.

In the limit of small $y = |\boldsymbol{y}|$ or large $\Delta = |\boldsymbol{\Delta}|$, renormalised DPDs can be computed in terms of renormalised PDFs and perturbative splitting kernels as

$$
F_{a_1a_2}(x_1, x_2, y, \mu) = \frac{1}{\pi y^2} \sum_{a_0} \int_{x_1+x_2}^1 \frac{dz}{z^2} V_{a_1a_2,a_0}\left(\frac{x_1}{z}, \frac{x_2}{z}, a_s(\mu), \log\frac{\mu^2 y^2}{b_0^2}\right) f_{a_0}(z, \mu),
$$
$$
F_{a_1a_2}(x_1, x_2, \Delta, \mu) = \sum_{a_0} \int_{x_1+x_2}^1 \frac{dz}{z^2} W_{a_1a_2,a_0}\left(\frac{x_1}{z}, \frac{x_2}{z}, a_s(\mu), \log\frac{\mu^2}{\Delta^2}\right) f_{a_0}(z, \mu), \tag{5}
$$

where $\mu$ is the renormalisation scale and we abbreviate

$$
a_s = \frac{\alpha_s}{2\pi}, \qquad\qquad b_0 = 2e^{-\gamma}, \tag{6}
$$

with $\gamma$ being the Euler-Mascheroni constant. The general form of the convolution integrals in (5) was already given in [43] and follows from boost invariance and dimensional analysis. The splitting mechanism expressed in (5) also induces an inhomogeneous term in the evolution equation for momentum space DPDs, which reads [54]

$$\frac{d}{d\ln\mu^2} F_{a_1 a_2}(x_1, x_2, \Delta, \mu) = \sum_{a_0} \int_{x_1+x_2}^{1} \frac{dz}{z^2} P_{a_1 a_2, a_0}\left(\frac{x_1}{z}, \frac{x_2}{z}, a_s(\mu)\right) f_{a_0}(z, \mu)$$
$$+ \{\text{homogeneous terms}\}, \tag{7}$$

where the homogeneous terms have the same form as in the evolution of $F_{a_1 a_2}(x_1, x_2, y, \mu)$ and are given in (22) below. At leading order (LO) in $a_s$, the kernel $P_{a_1 a_2, a_0}$ can be expressed in terms of the familiar LO DGLAP splitting functions, as specified in (37). The evolution equation (7) then takes the form that has been derived long ago [67,68]. Furthermore, one has $V_{a_1 a_2, a_0} = P_{a_1 a_2, a_0}$ at LO, as shown in [11]. In the present work we compute all kernels $V$, $W$ and $P$ at next-to-leading order (NLO) in $a_s$.

We refer to $V_{a_1 a_2, a_0}$ and $W_{a_1 a_2, a_0}$ as "$1 \to 2$ splitting kernels" and to $P_{a_1 a_2, a_0}$ as "$1 \to 2$ evolution kernel", where "$1 \to 2$" refers to having one initial parton $a_0$ that splits into the two partons $a_1, a_2$ specified in the DPD. Starting at NLO, the splitting process can involve further unobserved partons in the final state. At LO the flavour of $a_0$ is uniquely fixed for given $a_1, a_2$, so that the sum over $a_0$ can be omitted in (5) and (7), but this no longer holds at NLO.

Let us now introduce some elements of notation that will be useful for the calculations in the following sections.

**Dimensional regularisation.** Working in $d = 4 - 2\epsilon$ dimensions, we expand $\epsilon$ dependent quantities $Q(\epsilon)$ in a Laurent series around $\epsilon = 0$,

$$Q(\epsilon) = \sum_k \epsilon^k [Q]_k. \tag{8}$$

In particular, $[Q]_0$ gives the finite piece and $[Q]_{-1}$ the residue of the $1/\epsilon$ pole.

**Convolution integrals.** Following [54], we use a compact notation for the different types of convolution integrals that avoids giving explicit momentum fraction arguments. In the following, let $D$ be a function of $x_1, x_2$ whilst $A, B, C$ are functions of one momentum fraction only. We write

$$A \underset{1}{\otimes} D = \int_{x_1}^{1} \frac{dz}{z} A(z) D\left(\frac{x_1}{z}, x_2\right), \tag{9}$$

where the integration region is determined by the support properties of the functions. If a one-variable function $A$ is a PDF then $A(x) = 0$ for $x < 0$ and $x > 1$, and if a two-variable function $D$ is a DPD, then $D(x_1, x_2) = 0$ for $x_1 < 0$, $x_2 < 0$ and $x_1 + x_2 > 1$. Corresponding support properties hold for renormalisation factors and splitting kernels (which may include delta functions and plus distributions at the endpoints of their support). In analogy to (9), we define $A \underset{2}{\otimes} D$ and the combined convolution $A \underset{1}{\otimes} B \underset{2}{\otimes} D = A \underset{1}{\otimes} [B \underset{2}{\otimes} D]$.

In (5) and in (7) we encounter a second type of convolution,

$$D \underset{12}{\otimes} A = \int_{x_1+x_2}^{1} \frac{dz}{z^2} D\left(\frac{x_1}{z}, \frac{x_2}{z}\right) A(z). \tag{10}$$

Interchanging the order of integrations, it is straightforward to show that

$$[A \underset{1}{\otimes} D] \underset{12}{\otimes} B = A \underset{1}{\otimes} [D \underset{12}{\otimes} B], \qquad [A \underset{1}{\otimes} B \underset{2}{\otimes} D] \underset{12}{\otimes} C = A \underset{1}{\otimes} B \underset{2}{\otimes} [D \underset{12}{\otimes} C], \qquad (11)$$

so that we can write these convolutions without the brackets. We also have

$$[D \underset{12}{\otimes} A] \underset{12}{\otimes} B = D \underset{12}{\otimes} [A \otimes B], \qquad (12)$$

where $A \otimes B$ is the usual convolution for functions of a single momentum fraction. We will also write $D \underset{12}{\otimes} A \otimes B$ without brackets. One can rewrite (10) in the form

$$D \underset{12}{\otimes} A = \frac{1}{x_1 + x_2} \int_{x_1+x_2}^{1} dz \, D\left(z \frac{x_1}{x_1 + x_2}, z \frac{x_2}{x_1 + x_2}\right) A\left(\frac{x_1 + x_2}{z}\right), \qquad (13)$$

which will be useful in section 5.

**Inverse convolution (one variable).** We define $A^{-1}(x)$ as the solution of $A \otimes A^{-1} = \delta(1-x)$. We will only use this inversion for functions $A(x)$ that have a series expansion in $a_s$; one then obtains a unique inverse $A^{-1}(x)$ order by order in perturbation theory. In particular, if $A(x) = 1 + a_s A^{(1)}(x) + \mathcal{O}(a_s^2)$, then $A^{-1}(x) = 1 - a_s A^{(1)}(x) + \mathcal{O}(a_s^2)$.

**Parton indices.** The notation for convolutions can be extended to include appropriate summation over indices denoting the parton type (quarks, antiquarks, gluons). For quantities $A_{ab}(x), B_{ab}(x)$ and $f_a(x)$ depending on a single momentum fraction, we write as usual

$$[A \otimes B]_{ac} = \sum_b A_{ab} \otimes B_{bc}, \qquad [A \otimes f]_a = \sum_b A_{ab} \otimes f_b. \qquad (14)$$

Note that we do *not* use the summation convention for parton indices, i.e. sums are indicated explicitly when indices are given. For quantities $D_{a_1 a_2, a_0}(x_1, x_2)$ and $F_{a_1 a_2}(x_1, x_2)$ depending on two momentum fractions, we define

$$[A \underset{1}{\otimes} D]_{a_1 a_2, a_0} = \sum_b A_{a_1 b} \underset{1}{\otimes} D_{b a_2, a_0}, \qquad [A \underset{1}{\otimes} F]_{a_1 a_2} = \sum_b A_{a_1 b} \underset{1}{\otimes} F_{b a_2},$$

$$[A \underset{2}{\otimes} D]_{a_1 a_2, a_0} = \sum_b A_{a_2 b} \underset{1}{\otimes} D_{a_1 b, a_0}, \qquad [A \underset{2}{\otimes} F]_{a_1 a_2} = \sum_b A_{a_2 b} \underset{1}{\otimes} F_{a_1 b},$$

$$[D \underset{12}{\otimes} A]_{a_1 a_2, a_0} = \sum_b D_{a_1 a_2, b} \underset{12}{\otimes} A_{b a_0}, \qquad [D \underset{12}{\otimes} f]_{a_1 a_2} = \sum_b D_{a_1 a_2, b} \underset{12}{\otimes} f_b. \qquad (15)$$

For the combination of convolutions in the first and second momentum fraction, we then have

$$[A \underset{1}{\otimes} B \underset{2}{\otimes} D]_{a_1 a_2, a_0} = \sum_{b_1, b_2} A_{a_1 b_1} \underset{1}{\otimes} B_{a_2 b_2} \underset{2}{\otimes} D_{b_1 b_2, a_0}. \qquad (16)$$

## 2.2 Renormalisation and evolution of DPDs

Let us recall the renormalisation of DPDs, which is well-known at LO and has been formulated at arbitrary perturbative order in [54]. The twist-two operators in (2) contain short-distance singularities that are renormalised with a factor $Z(x, \mu)$ as[1]

$$f(\mu) = Z(\mu) \otimes f_B \qquad (17)$$

---

[1]Our convention for renormalisation factors $Z$ of composite operators corresponds to the one in [69,70]. Other authors, such as the ones of [71], use $Z^{-1}$ instead.

for ordinary PDFs. Correspondingly, $F_B(y)$ is renormalised by a factor $Z$ for each parton,

$$F(y,\mu) = Z(\mu) \underset{1}{\otimes} Z(\mu) \underset{2}{\otimes} F_B. \tag{18}$$

In addition to the divergences of the twist-two operators, the momentum-space distributions $F_B(\Delta)$ have a $1 \to 2$ splitting singularity, which arises from integrating $F_B(y)$ over $\boldsymbol{y}$ in (4) because $F_B(y) \propto y^{-2+2\epsilon}$ at small $y$. This singularity is renormalised additively with a factor $Z_s(x_1, x_2, \mu)$ as

$$F(\Delta,\mu) = Z(\mu) \underset{1}{\otimes} Z(\mu) \underset{2}{\otimes} F_B(\Delta) + Z_s(\mu) \underset{12}{\otimes} f_B. \tag{19}$$

Defining the ordinary DGLAP evolution kernel $P$ by

$$\frac{d}{d\log\mu^2} Z(\mu) = P(\mu) \otimes Z(\mu) \tag{20}$$

and using that bare matrix elements are $\mu$ independent, one obtains the usual DGLAP equations for $f(\mu)$ by taking the $\mu$ derivative of (17). In analogy, one obtains the homogeneous double DGLAP equation

$$\frac{d}{d\log\mu^2} F(y,\mu) = P(\mu) \underset{1}{\otimes} F(y,\mu) + P(\mu) \underset{2}{\otimes} F(y,\mu) \tag{21}$$

from (18). Taking the $\mu$ derivative of (19), one obtains the inhomogeneous double DGLAP equation

$$\frac{d}{d\log\mu^2} F(\Delta,\mu) = P(\mu) \underset{1}{\otimes} F(\Delta,\mu) + P(\mu) \underset{2}{\otimes} F(\Delta,\mu) + P_s(\mu) \underset{12}{\otimes} f(\mu), \tag{22}$$

with the $1 \to 2$ evolution kernel

$$P_s(\mu) = \left[ \left( \frac{d}{d\log\mu^2} Z_s(\mu) \right) - P(\mu) \underset{1}{\otimes} Z_s(\mu) - P(\mu) \underset{2}{\otimes} Z_s(\mu) \right] \underset{12}{\otimes} Z^{-1}(\mu), \tag{23}$$

where $Z^{-1}$ was defined in the previous subsection. Making parton indices explicit, we have DPDs $F_{a_1 a_2}$ and PDFs $f_a$. Correspondingly, the renormalisation factors and evolution kernels for PDFs read $Z_{ab}$ and $P_{ab}$ as usual. For the quantities associated with $1 \to 2$ splitting in DPDs we write $Z_{a_1 a_2, a_0}$ and $P_{a_1 a_2, a_0}$, omitting the subscript $s$ on $Z$ and $P$ when parton indices are written out.

## 2.3 Splitting kernels at higher order

Let us now see how renormalisation affects the $1 \to 2$ splitting kernels $V$ and $W$. For unrenormalised DPDs at small $y$ or large $\Delta$, the splitting mechanism gives factorisation formulae

$$F_B(y) = \frac{\Gamma(1-\epsilon)}{(\pi y^2)^{1-\epsilon}} V_B(y) \underset{12}{\otimes} f_B, \qquad\qquad F_B(\Delta) = W_B(\Delta) \underset{12}{\otimes} f_B, \tag{24}$$

with bare kernels $V_B$ and $W_B$, where the prefactor of $V_B(y)$ in $F_B(y)$ is taken out for convenience.[2] According to (17) to (19), we then have

$$F(y,\mu) = \frac{\Gamma(1-\epsilon)}{(\pi y^2)^{1-\epsilon}} V(y,\mu) \underset{12}{\otimes} f(\mu), \qquad\qquad F(\Delta,\mu) = W(\Delta,\mu) \underset{12}{\otimes} f(\mu), \tag{25}$$

with kernels

$$V(y,\mu) = Z(\mu) \underset{1}{\otimes} Z(\mu) \underset{2}{\otimes} V_B(y) \underset{12}{\otimes} Z^{-1}(\mu),$$

$$W(\Delta,\mu) = Z(\mu) \underset{1}{\otimes} Z(\mu) \underset{2}{\otimes} W_B(\Delta) \underset{12}{\otimes} Z^{-1}(\mu) + Z_s(\mu) \underset{12}{\otimes} Z^{-1}(\mu). \tag{26}$$

for renormalised distributions.

---

[2]Compared with [43], this prefactor includes an additional $\Gamma(1-\epsilon)$.

$\overline{\text{MS}}$ **implementation and coupling renormalisation.** For the bare momentum space kernel, we have a perturbative expansion

$$W_B(\Delta) = \sum_n \left(\frac{\alpha_0}{2\pi}\right)^n W_0^{(n)}(\Delta) = \sum_n \left(\frac{\mu^{2\epsilon}}{S_\epsilon} \frac{\alpha_s(\mu)}{2\pi} Z_\alpha(\mu)\right)^n W_0^{(n)}(\Delta)$$

$$= \sum_n a_s^n(\mu) \left(\frac{\mu}{\Delta}\right)^{2\epsilon n} Z_\alpha^n(\mu) W_B^{(n)}, \tag{27}$$

where $\alpha_0$ is the bare coupling. In the last step, we used the abbreviation (6) and introduced coefficients

$$W_B^{(n)}(x_1, x_2, \epsilon) = \Delta^{2\epsilon n} S_\epsilon^{-n} W_0^{(n)}(x_1, x_2, \Delta, \epsilon), \tag{28}$$

which are $\Delta$ independent, because $W_0^{(n)}(\Delta) \propto \Delta^{-2\epsilon n}$ for dimensional reasons. As discussed in [54], we implement the $\overline{\text{MS}}$ prescription by dividing bare graphs of order $n$ by $S_\epsilon^n$. The coupling renormalisation constant $Z_\alpha$ is then 1 plus a sum of pure poles in $\epsilon$. The standard choice in the literature is $S_\epsilon = (4\pi e^{-\gamma})^\epsilon$. An alternative choice $S_\epsilon = (4\pi)^\epsilon / \Gamma(1-\epsilon)$ was proposed by Collins [70], with the statement that for quantities for which ultraviolet divergences give at most one power of $\epsilon^{-1}$ per loop, the two schemes give identical results for renormalised quantities at $\epsilon = 0$. We shall verify this explicitly up to order $a_s^2$ in section 2.3.1.

Up to order $a_s$, we have

$$Z_\alpha = 1 - \frac{a_s}{\epsilon} \frac{\beta_0}{2} + \mathcal{O}(a_s^2), \qquad\qquad \beta_0 = \frac{11}{3} C_A - \frac{4}{3} T_F n_F, \tag{29}$$

where for brevity we do not display the $\mu$ dependence of $Z_\alpha$ and $a_s$. Here $T_F = 1/2$, and $n_F$ is the number of active quarks. This gives

$$\frac{da_s}{d\log\mu^2} = \frac{\beta(a_s)}{2\pi} = -\frac{\beta_0}{2} a_s^2 + \mathcal{O}(a_s^3) \tag{30}$$

for the running coupling in 4 dimensions, whereas the renormalisation group derivative in $4 - 2\epsilon$ dimensions reads

$$\frac{d}{d\log\mu^2} = \frac{\partial}{\partial\log\mu^2} + \left[\frac{\beta(a_s)}{2\pi} - \epsilon a_s\right] \frac{\partial}{\partial a_s}. \tag{31}$$

Expanding (27) up to order $a_s^2$, we obtain

$$W_B(\Delta) = a_s \left(\frac{\mu}{\Delta}\right)^{2\epsilon} W_B^{(1)} + a_s^2 \left(\frac{\mu}{\Delta}\right)^{4\epsilon} W_B^{(2)} - a_s^2 \left(\frac{\mu}{\Delta}\right)^{2\epsilon} \frac{\beta_0}{2\epsilon} W_B^{(1)} + \mathcal{O}(a_s^3) \tag{32}$$

for the bare $1 \to 2$ splitting kernel.

**Renormalisation factors and splitting functions.** The perturbative expansion of the renormalisation functions $Z$ and $Z_s$ reads

$$Z(x, \epsilon, \mu) = \delta(1-x) + \sum_{n=1}^{\infty} a_s^n(\mu) Z^{(n)}(x, \epsilon),$$

$$Z_s(x_1, x_2, \epsilon, \mu) = \sum_{n=1}^{\infty} a_s^n(\mu) Z_s^{(n)}(x_1, x_2, \epsilon). \tag{33}$$

Note that $Z_s$ has no tree level term. With the implementation of the $\overline{\text{MS}}$ scheme discussed below (27), all counterterms are pure poles in $\epsilon$, and we have

$$\left[Z^{(1)}\right]_{-1} = -P^{(0)}, \qquad \left[Z_s^{(1)}\right]_{-1} = -P_s^{(0)}, \qquad \left[Z_s^{(2)}\right]_{-1} = -P_s^{(1)}/2, \qquad (34)$$

where the evolution kernels are expanded as

$$P(x,\mu) = \sum_{n=0}^{\infty} a_s^{n+1}(\mu) P^{(n)}(x), \qquad P_s(x_1,x_2,\mu) = \sum_{n=0}^{\infty} a_s^{n+1}(\mu) P_s^{(n)}(x_1,x_2). \qquad (35)$$

The residues of $1/\epsilon$ in (34) are obtained by inserting (31) into (20) and (23) and then comparing the $\mathcal{O}(\epsilon^0)$ terms on both sides, taking into account that the evolution kernels are independent of $\epsilon$, see e.g. [54]. Comparing the $\mathcal{O}(\epsilon^{-1})$ terms, one finds that $[Z]_{-2}$ and $[Z_s]_{-2}$ start at order $a_s^2$. In particular, one has

$$\left[Z_s^{(2)}\right]_{-2} = \frac{1}{2}\left(\frac{\beta_0}{2}P_s^{(0)} + P^{(0)}\underset{1}{\otimes}P_s^{(0)} + P^{(0)}\underset{2}{\otimes}P_s^{(0)} + P_s^{(0)}\underset{12}{\otimes}P^{(0)}\right). \qquad (36)$$

At LO, the inhomogeneous evolution kernel has the form

$$P_s^{(0)}(x_1,x_2) = \delta(1-x_1-x_2) P_s^{(0)}(x_1) \qquad (37)$$

with a kinematic constraint due to the fact that the splitting process gives exactly two final state partons at this order. The one-variable kernel $P_s(x)$ is equal to $P^{(0)}(x)$, except that it has no $\delta(1-x)$ terms and no plus prescription on $1/(1-x)$ factors. The convolutions in (36) hence turn into ordinary products:

$$P^{(0)}\underset{1}{\otimes}P_s^{(0)} = P^{(0)}\left(\frac{x_1}{1-x_2}\right)\frac{P_s^{(0)}(1-x_2)}{1-x_2},$$
$$P^{(0)}\underset{2}{\otimes}P_s^{(0)} = P^{(0)}\left(\frac{x_2}{1-x_1}\right)\frac{P_s^{(0)}(x_1)}{1-x_1},$$
$$P_s^{(0)}\underset{12}{\otimes}P^{(0)} = P_s^{(0)}\left(\frac{x_1}{x_1+x_2}\right)\frac{P^{(0)}(x_1+x_2)}{x_1+x_2}. \qquad (38)$$

Note that for flavour diagonal transitions these expressions still contain distributions, which will be made explicit in (135) and (136).

### 2.3.1 Momentum space kernels

We expand the renormalised kernel (26) in $4-2\epsilon$ dimensions as

$$W(\Delta,\mu,\epsilon) = \sum_{n=1}^{\infty} a_s^n(\mu) W^{(n)}(\Delta,\mu,\epsilon), \qquad W^{(n)}(\Delta,\mu,\epsilon) = \sum_{m=0}^{n}\left(\frac{\mu}{\Delta}\right)^{2\epsilon m} W^{(n,m)}(\epsilon). \qquad (39)$$

The coefficient $W^{(n,n)}$ equals the $n$th order term $W_B^{(n)}$ of the bare kernel $W_B$ in (27), the coefficients $W^{(n,m)}$ with $0 < m < n$ are products of lower order terms $W_B^{(m)}$ with renormalisation counterterms for the twist-two operators or the QCD coupling, whilst $W^{(n,0)}$ comes from the counterterm for the splitting singularity. Expanding $(\mu/\Delta)^{2\epsilon m}$ in $\epsilon$, we obtain the kernel in 4 dimensions,

$$W^{(n)}(\Delta,\mu) \underset{\text{def}}{=} W^{(n)}(\Delta,\mu,\epsilon=0) = \sum_{k=0}^{n}\left(\log\frac{\mu^2}{\Delta^2}\right)^k W^{[n,k]}, \qquad (40)$$

with

$$W^{[n,0]} = \sum_{m=0}^{n} \left[ W^{(n,m)} \right]_0, \qquad W^{[n,k]} = \sum_{m=1}^{n} \frac{m^k}{k!} \left[ W^{(n,m)} \right]_{-k} \text{ for } k \geq 1. \qquad (41)$$

The absence of poles $1/\epsilon^j$ in $W^{(n)}$ yields the finiteness conditions

$$0 = \sum_{m=0}^{n} \left[ W^{(n,m)} \right]_{-j}, \qquad 0 = \sum_{m=1}^{n} m^k \left[ W^{(n,m)} \right]_{-k-j} \text{ for } k \geq 1, \qquad (42)$$

where $j \geq 1$ in all cases.

At LO we have

$$W^{(1,1)} = W_B^{(1)}, \qquad W^{(1,0)} = Z_s^{(1)} = -\epsilon^{-1} P_s^{(0)}, \qquad (43)$$

where in the last step we used (34). $W^{(1)}$ has at most single poles, and the first condition in (42) implies

$$\left[ W_B^{(1)} \right]_{-1} = P_s^{(0)}. \qquad (44)$$

The coefficients of the kernel (40) in 4 dimensions thus read

$$W^{[1,0]} = \left[ W^{(1,1)} \right]_0 = \left[ W_B^{(1)} \right]_0, \qquad W^{[1,1]} = \left[ W^{(1,1)} \right]_{-1} = P_s^{(0)}. \qquad (45)$$

At NLO we have

$$W^{(2,2)} = W_B^{(2)},$$

$$W^{(2,1)} = -\frac{\beta_0}{2\epsilon} W_B^{(1)} + Z^{(1)} \underset{1}{\otimes} W_B^{(1)} + Z^{(1)} \underset{2}{\otimes} W_B^{(1)} - W_B^{(1)} \underset{12}{\otimes} Z^{(1)}$$

$$= -\frac{1}{\epsilon} \left( \frac{\beta_0}{2} W_B^{(1)} + P^{(0)} \underset{1}{\otimes} W_B^{(1)} + P^{(0)} \underset{2}{\otimes} W_B^{(1)} - W_B^{(1)} \underset{12}{\otimes} P^{(0)} \right),$$

$$W^{(2,0)} = Z_s^{(2)} - Z_s^{(1)} \underset{12}{\otimes} Z^{(1)}$$

$$= -\frac{1}{2\epsilon} P_s^{(1)} + \frac{1}{\epsilon^2} \left[ Z_s^{(2)} \right]_{-2} - \frac{1}{\epsilon^2} P_s^{(0)} \underset{12}{\otimes} P^{(0)}, \qquad (46)$$

where in the second steps we used (34) again. The finiteness conditions (42) now read

$$0 = \left[ W^{(2,2)} \right]_{-1} + \left[ W^{(2,1)} \right]_{-1} + \left[ W^{(2,0)} \right]_{-1},$$
$$0 = \left[ W^{(2,2)} \right]_{-2} + \left[ W^{(2,1)} \right]_{-2} + \left[ W^{(2,0)} \right]_{-2},$$
$$0 = 2 \left[ W^{(2,2)} \right]_{-2} + \left[ W^{(2,1)} \right]_{-2}. \qquad (47)$$

Using (36) and the fact the double poles in $W^{(2,1)}$ are obtained by replacing $W_B^{(1)}$ with $\epsilon^{-1} P_s^{(0)}$, we get

$$\left[ W_B^{(2)} \right]_{-2} = \frac{1}{2} \left( \frac{\beta_0}{2} P_s^{(0)} + P^{(0)} \underset{1}{\otimes} P_s^{(0)} + P^{(0)} \underset{2}{\otimes} P_s^{(0)} - P_s^{(0)} \underset{12}{\otimes} P^{(0)} \right). \qquad (48)$$

This relation fixes the double pole of $W_B^{(2)}$ in terms of the LO kernels and thus serves as a cross check of the two-loop calculation. For the single poles, we obtain

$$\left[ W_B^{(2)} \right]_{-1} = \frac{1}{2} P_s^{(1)} + \left( \frac{\beta_0}{2} \left[ W_B^{(1)} \right]_0 + P^{(0)} \underset{1}{\otimes} \left[ W_B^{(1)} \right]_0 + P^{(0)} \underset{2}{\otimes} \left[ W_B^{(1)} \right]_0 - \left[ W_B^{(1)} \right]_0 \underset{12}{\otimes} P^{(0)} \right), \quad (49)$$

from which we can extract the NLO evolution kernel $P_s^{(1)}$. Collecting the finite contributions, we get

$$W^{[2,0]} = \left[W_B^{(2)}\right]_0 - \left(\frac{\beta_0}{2}\left[W_B^{(1)}\right]_1 + P^{(0)}\underset{1}{\otimes}\left[W_B^{(1)}\right]_1 + P^{(0)}\underset{2}{\otimes}\left[W_B^{(1)}\right]_1 - \left[W_B^{(1)}\right]_1\underset{12}{\otimes}P^{(0)}\right),$$

$$W^{[2,1]} = P_s^{(1)} + \frac{\beta_0}{2}W^{[1,0]} + P^{(0)}\underset{1}{\otimes}W^{[1,0]} + P^{(0)}\underset{2}{\otimes}W^{[1,0]} - W^{[1,0]}\underset{12}{\otimes}P^{(0)},$$

$$W^{[2,2]} = \frac{1}{2}\left(\frac{\beta_0}{2}P_s^{(0)} + P^{(0)}\underset{1}{\otimes}P_s^{(0)} + P^{(0)}\underset{2}{\otimes}P_s^{(0)} - P_s^{(0)}\underset{12}{\otimes}P^{(0)}\right), \tag{50}$$

where we used (45) to replace $\left[W_B^{(1)}\right]_0$ by $W^{[1,0]}$. Inserting (45) and (50) into the factorisation formula (25) for $F(\Delta,\mu)$ and taking the $\mu$ derivative, one obtains the inhomogeneous double DGLAP equation (22) up to order $a_s^2$ as a cross check.

To evaluate (50) one needs $W_B^{(1)}$ up to $\mathcal{O}(\epsilon)$. The full expressions for $W_B^{(1)}$ in $d = 4-2\epsilon$ dimensions are easily computed along the lines of section 5 in [11] and read

$$W_B^{(1)}(x_1, x_2) = \delta(1-x_1-x_2)W_B^{(1)}(x_1), \tag{51}$$

with

$$\epsilon R_\epsilon W_{B,gg,g}^{(1)}(x) = 2C_A\left[\frac{x}{1-x} + \frac{1-x}{x} + x(1-x)\right],$$

$$\epsilon R_\epsilon W_{B,q\bar{q},g}^{(1)}(x) = T_F\frac{x^2 + (1-x)^2 - \epsilon}{1-\epsilon},$$

$$\epsilon R_\epsilon W_{B,qg,q}^{(1)}(x) = C_F\left[\frac{1+x^2}{1-x} - \epsilon(1-x)\right]. \tag{52}$$

Here we introduced the factor

$$R_\epsilon = \frac{S_\epsilon}{(4\pi)^\epsilon}\frac{\Gamma(1-2\epsilon)}{\Gamma(1+\epsilon)\Gamma^2(1-\epsilon)} = 1 + \mathcal{O}(\epsilon^2), \tag{53}$$

with $S_\epsilon$ specified below (28). Since $W_B^{(1)}(x_1, x_2)$ has the same kinematic constraint on $x_1$ and $x_2$ as $P_s^{(0)}$ in (37), the convolutions in (50) are products as in (38), with $P_s^{(0)}$ replaced as appropriate.

The square of the tree-level graphs for the splitting $a_0 \to a_1 a_2$ that give rise to (52) appears in many higher-order calculations and has been computed in many papers before. Note however that $W_B$ is computed for transverse momenta of $a_1$ and $a_2$ that differ by $\pm\Delta$ between the amplitude and its conjugate (see section 4.1). This is typically not the case in other contexts. Nevertheless, the expressions on the r.h.s. of (52) agree with the expressions for $P^{n\neq 4}$ given in section 3 of [72].

**Equivalence of $\overline{\text{MS}}$ scheme implementations.** Let us show that the two choices for $S_\epsilon$ in the implementation of the $\overline{\text{MS}}$ scheme specified below (28) give the same kernels $W^{[n,k]}$ in 4 dimensions at LO and NLO. The choices coincide in the constant and the first-order term $[S_\epsilon]_1$ of the Taylor expansion of $S_\epsilon$ around $\epsilon = 0$, but they differ in the second-order term $[S_\epsilon]_2$. Because $W_B^{(1)}$ has only a single pole in $\epsilon$, the renormalised LO kernels are identical for the two choices, as are the associated renormalisation factors $Z^{(1)}$ and $Z_s^{(1)}$. Since $W_B^{(2)}$ contains at most double poles in $\epsilon$, the NLO renormalisation factor $Z_s^{(2)}$ is the same for the two choices as well. The only remaining dependence on $[S_\epsilon]_2$ in the renormalised two loop kernel $W^{(2)}(\Delta,\mu)$

at $\epsilon = 0$ can thus come from the terms

$$
\begin{aligned}
&\left[W^{(2,2)}\right]_{-2}\left[S_\epsilon^{-2}\right]_2 + \left[W^{(2,1)}\right]_{-2}\left[S_\epsilon^{-1}\right]_2 \\
&= \left(3\left[W^{(2,2)}\right]_{-2} + \left[W^{(2,1)}\right]_{-2}\right)\left(\left[S_\epsilon\right]_1\right)^2 - \left(2\left[W^{(2,2)}\right]_{-2} + \left[W^{(2,1)}\right]_{-2}\right)\left[S_\epsilon\right]_2,
\end{aligned}
\tag{54}
$$

where the dependence on $S_\epsilon$ follows from (28) and is due to the terms $W_B^{(2)}$ and $W_B^{(1)}$ in (46). Thanks to the last relation in (47), the dependence on $\left[S_\epsilon\right]_2$ cancels in this expression, which completes our argument.

### 2.3.2 Position space kernels

The bare kernels $V_B$ and $W_B$ are related by a Fourier transform according to (4). Using equation (E.1) in [73], one gets for the $a_s^n$ term in (27)

$$
\int \frac{d^{2-2\epsilon}\Delta}{(2\pi)^{2-2\epsilon}}\, e^{-i\Delta y}\left(\frac{\mu}{\Delta}\right)^{2\epsilon n} W_B^{(n)} = \frac{\Gamma(1-\epsilon)}{(\pi y^2)^{1-\epsilon}}\left(\frac{y\mu}{b_0}\right)^{2\epsilon n} n\epsilon\, T_{\epsilon,n}\, W_B^{(n)},
\tag{55}
$$

with $b_0$ given in (6) and

$$
T_{\epsilon,n} = \frac{\Gamma(1-\epsilon-\epsilon n)}{\Gamma(1+\epsilon n)\Gamma(1-\epsilon)}\, e^{-2n\gamma\epsilon} = 1 + \zeta_2\, n\epsilon^2 + \frac{\zeta_3}{3}\left(2n^3 + 3n^2 + 3n\right)\epsilon^3 + \mathcal{O}(\epsilon^4),
\tag{56}
$$

where $\zeta_n$ denotes the Riemann $\zeta$ function evaluated at integer argument $n$. We define[3]

$$
V(y,\mu,\epsilon) = \sum_{n=1}^{\infty} a_s^n(\mu)\, V^{(n)}(y,\mu,\epsilon), \qquad V^{(n)}(y,\mu,\epsilon) = \sum_{m=1}^{n}\left(\frac{y\mu}{b_0}\right)^{2\epsilon m} V^{(n,m)}(\epsilon).
\tag{57}
$$

In contrast to (39) there is no $m = 0$ term here, because there is no splitting singularity in $y$ space. By contrast, the counterterms for the twist-two operators and the QCD coupling are common to momentum and position space. One thus has

$$
V^{(n,m)}(\epsilon) = m\epsilon\, T_{\epsilon,m}\, W^{(n,m)}(\epsilon).
\tag{58}
$$

For the kernel at $\epsilon = 0$ we write

$$
V^{(n)}(y,\mu) \underset{\text{def}}{=} V^{(n)}(y,\mu,\epsilon=0) = \sum_{k=0}^{n-1}\left(\log\frac{y^2\mu^2}{b_0^2}\right)^k V^{[n,k]}.
\tag{59}
$$

The coefficients $V^{[n,k]}$ and $V^{(n,m)}$ obey exactly the same relations as $W^{[n,k]}$ and $W^{(n,m)}$ in (41) and (42), except that the sum over $m$ always starts at $m = 1$.

At LO and NLO, the relation (58) implies

$$
\left[V^{(n,m)}\right]_{-k} = \left[m\epsilon\, W^{(n,m)}\right]_{-k} = m\left[W^{(n,m)}\right]_{-k-1} \qquad \text{for } k \geq 0 \text{ and } n = 1,2.
\tag{60}
$$

The kernel $V^{(1)}$ is finite and $V^{(2)}$ has at most single poles, so that one has only one finiteness condition,

$$
0 = \left[V^{(2,2)}\right]_{-1} + \left[V^{(2,1)}\right]_{-1}.
\tag{61}
$$

---

[3] Here and in (59) we use a convention that differs from equations (3.15) and (3.16) in [43].

This is satisfied thanks to (60) for $k = 1$ and $n = 2$, together with the third relation in (47). For the finite part of the kernel, the relations (60) imply

$$V^{[n,k-1]} = \sum_{m=1}^{n} \frac{m^k}{(k-1)!} \big[ W^{(n,m)} \big]_{-k} = k \sum_{m=0}^{n} \frac{m^k}{k!} \big[ W^{(n,m)} \big]_{-k}$$

$$= k\, W^{[n,k]} \qquad\qquad \text{for } 1 \le k \le n \text{ and } n = 1, 2, \tag{62}$$

where we used the relation (41) and its analogue for $V^{[n,k]}$. This gives the following simple relation between the kernels in 4 dimensions:

$$V^{(n)}(y,\mu) = \frac{\partial}{\partial \log \mu^2} W^{(n)}(\Delta = b_0/y, \mu) \qquad\qquad \text{for } n = 1, 2. \tag{63}$$

Explicitly, we have

$$V^{[1,0]} = W^{[1,1]} = P_s^{(0)},$$

$$V^{[2,0]} = W^{[2,1]} = P_s^{(1)} + \frac{\beta_0}{2} \big[ W_B^{(1)} \big]_0 + P^{(0)} \underset{1}{\otimes} \big[ W_B^{(1)} \big]_0 + P^{(0)} \underset{2}{\otimes} \big[ W_B^{(1)} \big]_0 - \big[ W_B^{(1)} \big]_0 \underset{12}{\otimes} P^{(0)},$$

$$V^{[2,1]} = 2W^{[2,2]} = \frac{\beta_0}{2} P_s^{(0)} + P^{(0)} \underset{1}{\otimes} P_s^{(0)} + P^{(0)} \underset{2}{\otimes} P_s^{(0)} - P_s^{(0)} \underset{12}{\otimes} P^{(0)}. \tag{64}$$

Inserting this into the factorisation formula (25) for $F(y, \mu)$ and taking the $\mu$ derivative, one correctly obtains the homogeneous double DGLAP equation (21) up to order $a_s^2$.

**Higher orders.** One may wonder whether (62) and thus (63) holds at all orders $n$. Consider therefore $n = 3$, where the $\mathcal{O}(\epsilon^2)$ term of $T_{\epsilon,n}$ contributes for the first time, namely to the finite part of $V^{(3)}$ but not to its poles. With (56) and (60) one has

$$V^{[3,0]} = \sum_{m=1}^{3} \big[ V^{(3,m)} \big]_0 = \sum_{m=1}^{3} m \big[ W^{(3,m)} \big]_{-1} + \zeta_2 \sum_{m=1}^{3} m^2 \big[ W^{(3,m)} \big]_{-3}. \tag{65}$$

The sum multiplying $\zeta_2$ is zero due to the constraint (42) for $k = 2, j = 1$. We thus find that (62) remains valid for $n = 3$. One order higher, we have

$$V^{[4,0]} = \sum_{m=1}^{4} \big[ V^{(4,m)} \big]_0 = \sum_{m=1}^{4} m \big[ W^{(4,m)} \big]_{-1} + \zeta_2 \sum_{m=1}^{4} m^2 \big[ W^{(4,m)} \big]_{-3}$$

$$+ \frac{\zeta_3}{3} \sum_{m=1}^{4} (2m^4 + 3m^2 + 3m) \big[ W^{(4,m)} \big]_{-4}. \tag{66}$$

According to (42), the sum over $m^k \big[ W^{(4,m)} \big]_{-4}$ is zero for $k = 1, 2, 3$ but not for $k = 4$. With (41) we get

$$V^{[4,0]} = W^{[4,1]} + 16\zeta_3 W^{[4,4]}, \tag{67}$$

and thus find that (62) is no longer valid for $n = 4$.

## 3 Matching between momentum and position space DPDs

In the formalism developed in [43], DPS cross sections are calculated using position space DPDs. On the other hand, one of the few theoretical constraints we have on the form of DPDs

is given by the sum rules introduced in [53], which are formulated for momentum space DPDs at $\boldsymbol{\Delta} = \mathbf{0}$. Due to the splitting singularity, position and momentum space DPDs are defined with a different ultraviolet renormalisation. As a consequence, they are not simply related by a Fourier transform but by a short-distance matching formula, which can be calculated in perturbation theory and was derived at LO in [43]. We now generalise this matching to higher orders in $a_s$.

Consider a cut-off regularised momentum space DPD as defined in [43]:

$$F_{\Phi, a_1 a_2}(x_1, x_2, \boldsymbol{\Delta}, \mu, \nu) = \int d^2 \boldsymbol{y} \, e^{i y \boldsymbol{\Delta}} \, \Phi(y \, \nu) \, F_{a_1 a_2}(x_1, x_2, \boldsymbol{y}, \mu), \tag{68}$$

where $\Phi$ is a function satisfying $\Phi(u) \to 1$ for $u \to \infty$ and $\Phi(u) = O(u^{1+\delta})$ for $u \to 0$ with some $\delta > 0$. This function regulates the splitting divergence at small $y$ that one would encounter when naively Fourier transforming the physical position space DPD in $d = 4$ dimensions. Let us first consider the case where $F_\Phi$ is defined with a hard cut-off $\Phi(u) = \Theta(u - b_0)$. The difference between the $\overline{\text{MS}}$ renormalised momentum space DPD (19) and (68) is then given by

$$\begin{aligned}
F(\boldsymbol{\Delta}, \mu) - F_\Phi(\boldsymbol{\Delta}, \mu, \nu) &= \lim_{\epsilon \to 0} \left[ \int_{y < b_0/\nu} d^{2-2\epsilon} \boldsymbol{y} \, e^{i \boldsymbol{\Delta} \boldsymbol{y}} F(y, \epsilon) + Z_s(\epsilon) \underset{12}{\otimes} Z^{-1}(\epsilon) \otimes f \right] \\
&= \lim_{\epsilon \to 0} \left[ \int_{y < b_0/\nu} d^{2-2\epsilon} \boldsymbol{y} \, F(y, \epsilon) + Z_s(\epsilon) \underset{12}{\otimes} Z^{-1}(\epsilon) \otimes f \right] + \int_{y < b_0/\nu} d^2 \boldsymbol{y} \left[ e^{i \boldsymbol{\Delta} \boldsymbol{y}} - 1 \right] F(y).
\end{aligned} \tag{69}$$

Here we have explicitly written out the splitting counterterm from (19) and indicated all $\epsilon$ dependence, using the shorthand notation $F(y) = F(y, \epsilon = 0)$ in the last term. For brevity, the dependence of distributions and $Z$ factors on $\mu$ is not shown. The relation (69) allows us to separate the matching into a part at $\boldsymbol{\Delta} = \mathbf{0}$ that involves DPDs in $4 - 2\epsilon$ dimensions and a $\boldsymbol{\Delta}$ dependent part that involves only DPDs at the physical point $\epsilon = 0$.

In following is understood that the cut-off scale $\nu$ is large enough to justify replacing $F(y)$ with the perturbative splitting contribution (25), up to corrections in powers of $\Lambda / \nu$. As discussed in section 3.3 of [43], the corrections of order $\Lambda / \nu$ arise from twist-three distributions, which are expected to be small at low $x_1$ and $x_2$. The dominant power corrections are then of order $\Lambda^2 / \nu^2$.

We now turn to the case where $\Phi$ is not a hard cut-off. The difference between two momentum space DPDs defined with different regulator functions $\Phi_1$ and $\Phi_2$ is given by

$$F_{\Phi_1} - F_{\Phi_2} = \int d^2 \boldsymbol{y} \, e^{i \boldsymbol{\Delta} \boldsymbol{y}} \left[ \Phi_1(y \, \nu) - \Phi_2(y \, \nu) \right] F(y). \tag{70}$$

This involves only $F(y)$ at the physical point, because each regulator $\Phi_i$ ensures that integrals over $y$ are finite in the ultraviolet. For distances $y$ of hadronic size, the expression in square brackets vanishes because $\Phi_i \to 1$ in that limit, so that one can replace $F(y)$ by its perturbative splitting form (25). Using (59) one can thus express the difference (70) in terms of $V^{[n,k]} \underset{12}{\otimes} f(\mu)$ and the integrals

$$\int \frac{d(y^2)}{y^2} J_0(y \Delta) \left( \log \frac{y^2 \mu^2}{b_0^2} \right)^k \left[ \Phi_1(y \, \nu) - \Phi_2(y \, \nu) \right], \tag{71}$$

where the angular integration has been performed and has given rise to the Bessel function $J_0$. It is thus straightforward to perform a conversion between DPDs $F_{\Phi_i}$ defined with different regulator functions $\Phi_i$. In the following we will therefore limit ourselves to the choice $\Phi(u) = \Theta(u - b_0)$.

### 3.1 Matching at zero $\Delta$

Let us first discuss matching at $\Delta = 0$, which we wish to write as

$$F(\Delta = 0, \mu) - F_\Phi(\Delta = 0, \mu, \nu) = U(\mu, \nu) \underset{12}{\otimes} f(\mu) + \mathcal{O}(\Lambda/\nu). \tag{72}$$

To compute the matching kernel $U(x_1, x_2, \mu, \nu)$, we replace $F(y, \epsilon)$ in the first term on the r.h.s. of (69) by its perturbative splitting form (25). With the $y$ dependence in (57), this gives rise to the integrals

$$\int_{y<b_0/\nu} d^{2-2\epsilon}y \, \frac{\Gamma(1-\epsilon)}{(\pi y^2)^{1-\epsilon}} \left(\frac{y\mu}{b_0}\right)^{2\epsilon m} = \frac{1}{m\epsilon} \left(\frac{\mu}{\nu}\right)^{2\epsilon m}. \tag{73}$$

Expanding $U(x_1, x_2, \mu, \nu, \epsilon)$ as

$$U(\mu, \nu, \epsilon) = \sum_{n=1}^{\infty} a_s^n(\mu) U^{(n)}(\mu, \nu, \epsilon), \qquad U^{(n)}(\mu, \nu, \epsilon) = \sum_{m=0}^{n} \left(\frac{\mu}{\nu}\right)^{2\epsilon m} U^{(n,m)}(\epsilon) \tag{74}$$

in analogy to (57), we thus find that the coefficients $U^{(n,m)}$, $V^{(n,m)}$, and $W^{(n,m)}$ are related as

$$U^{(n,m)} = \frac{1}{m\epsilon} V^{(n,m)} = T_{\epsilon,m} W^{(n,m)} \qquad \text{for } m \geq 1, \tag{75}$$

with $T_{\epsilon,m}$ from (56), where in the last step we have used the relation (58) between $V^{(n,m)}$ and $W^{(n,m)}$. The $m = 0$ coefficient originates from the splitting counterterm in (69) and thus reads

$$U^{(n,0)} = W^{(n,0)}. \tag{76}$$

In physical dimensions, i.e. at $\epsilon = 0$, the matching kernel can again be written as

$$U^{(n)}(\mu, \nu) \underset{\text{def}}{=} U^{(n)}(\mu, \nu, \epsilon = 0) = \sum_{m=0}^{n} \left(\log \frac{\mu^2}{\nu^2}\right)^m U^{[n,m]}, \tag{77}$$

where $U^{[n,k]}$ and $U^{(n,m)}$ obey the same relations as $W^{[n,k]}$ and $W^{(n,m)}$ in (41). The coefficients $U^{(n,m)}$ fulfil finiteness relations analogous to those for $W^{(n,m)}$ in (42).

Since $T_{\epsilon,m} = 1 + \mathcal{O}(\epsilon^2)$, the pole terms of $U^{(n)}$ and $W^{(n)}$ coincide at LO and NLO, which ensures the validity of the finiteness conditions for $U$ and fixes all coefficients of the logarithms in (77). For the nonlogarithmic term we obtain

$$U^{[n,0]} = \sum_{m=0}^{n} \left[U^{(n,m)}\right]_0 = \sum_{m=0}^{n} \left[W^{(n,m)}\right]_0 + \zeta_2 \sum_{m=1}^{n} m \left[W^{(n,m)}\right]_{-2} \quad \text{for } n = 1, 2, \tag{78}$$

where in the second step we used (75), (76) and (56). The sum multiplying $\zeta_2$ vanishes due to the constraint (42) for $k = 1, j = 1$. In total, we thus have

$$U^{[n,k]} = W^{[n,k]} \qquad \text{for } 0 \leq k \leq n \text{ and } n = 1, 2. \tag{79}$$

**Higher orders.** Let us see whether the relation (79) remains valid at higher orders. To this end, we compute the coefficient

$$\begin{aligned}
U^{[3,0]} &= \sum_{m=0}^{3} \left[U^{(3,m)}\right]_0 \\
&= \sum_{m=0}^{3} \left[W^{(3,m)}\right]_0 + \zeta_2 \sum_{m=1}^{3} m \left[W^{(3,m)}\right]_{-2} + \frac{\zeta_3}{3} \sum_{m=1}^{3} (2m^3 + 3m + 3) \left[W^{(3,m)}\right]_{-3} \\
&= W^{[3,0]} + 4\zeta_3 W^{[3,3]},
\end{aligned} \tag{80}$$

where we have used (75) and (42). We thus find that (79) is no longer valid for $n \geq 3$. We can, however, derive from (75) a relation between the matching kernels and the position space kernels that is valid to all orders, namely

$$\left[U^{(n,m)}\right]_{-k-1} = \frac{1}{m}\left[V^{(n,m)}\right]_{-k} \qquad\qquad \text{for } k \geq 0, \ m \geq 1, \ \text{and all } n. \tag{81}$$

In analogy to (62), we thus get

$$U^{[n,k]} = \frac{1}{k} V^{[n,k-1]} \qquad\qquad \text{for } 1 \leq k \leq n \ \text{and all } n, \tag{82}$$

which for $n = 1, 2$ is of course consistent with (62) and (79).

## 3.2 Matching at nonzero $\Delta$

For matching at nonzero $\Delta$, we need both terms on the r.h.s. of (69). Using the perturbative splitting form of $F(y)$ (25) and the expansion (59) of the kernel $V$, one readily obtains

$$F(\Delta, \mu) - F_\Phi(\Delta, \mu, \nu) = U(\mu, \nu) \underset{12}{\otimes} f(\mu) + \sum_{n=1}^{\infty} a_s^n(\mu) \sum_{k=0}^{n-1} I_k(\Delta, \nu, \mu) V^{[n,k]} \otimes f(\mu), \tag{83}$$

with corrections of $\mathcal{O}(\Lambda/\nu)$ and coefficients $I_k$ given by the integrals

$$I_k(\Delta, \nu, \mu) = \int_0^{b_0^2/\nu^2} \frac{d(y^2)}{y^2}\left[J_0(y\Delta) - 1\right]\left(\log\frac{y^2\mu^2}{b_0^2}\right)^k, \tag{84}$$

which arise after performing the angular part of the $y$ integration. We make the $\mu$ dependence explicit by writing this as

$$I_k(\Delta, \mu, \nu) = \sum_{j=0}^{k}\binom{k}{j} I_j(\Delta, \nu, \nu)\left(\log\frac{\mu^2}{\nu^2}\right)^j, \tag{85}$$

where

$$I_j(\Delta, \nu, \nu) = 2\int_0^a \frac{dz}{z}\left[J_0(z) - 1\right]\left(\log\frac{z^2}{a^2}\right)^j = \frac{1}{j+1}\int_0^a dz\, J_1(z)\left(\log\frac{z^2}{a^2}\right)^{j+1}, \tag{86}$$

with $a = b_0\Delta/\nu$. In the limits $\Delta \ll \nu$ and $\nu \ll \Delta$, the above equations can be further simplified. In the former case the limiting form of (85) is easily obtained by Taylor expanding the Bessel function, yielding

$$I_k(\Delta, \nu, \mu) \underset{\Delta \ll \nu}{=} -\frac{b_0^2\Delta^2}{4\mu^2}\int_0^{\mu^2/\nu^2} dz\, \log^k z = -\frac{b_0^2\Delta^2}{4\nu^2}\left[\left(\log\frac{\mu^2}{\nu^2}\right)^k + \sum_{j=0}^{k-1}c_j\left(\log\frac{\mu^2}{\nu^2}\right)^j\right], \tag{87}$$

with numerical coefficients $c_j$. The limiting behaviour for $\nu \ll \Delta$ is obtained by writing out the polynomial series of $\log^{k+1}(z^2/a^2) = \left[\log(\nu^2/\Delta^2) + \log(z^2/b_0^2)\right]^{k+1}$ and extending the $z$ integration to infinity in (86). This gives

$$I_j(\Delta, \nu, \nu) \underset{\nu \ll \Delta}{=} \frac{1}{j+1}\left[\left(\log\frac{\nu^2}{\Delta^2}\right)^{j+1} + \sum_{i=0}^{j}d_i\left(\log\frac{\nu^2}{\Delta^2}\right)^i\right], \tag{88}$$

with numerical coefficients $d_i$. Plugging this back into (85) and rearranging the binomial sum, we obtain the following expression for the large $\Delta$ behaviour:

$$I_k(\Delta, \nu, \mu) \underset{\nu \ll \Delta}{=} \frac{1}{k+1}\left[\left(\log \frac{\mu^2}{\Delta^2}\right)^{k+1} - \left(\log \frac{\mu^2}{\nu^2}\right)^{k+1}\right] + \cdots, \tag{89}$$

where the ellipsis denotes terms with fewer powers of logarithms.

To use the matching equation (83) truncated in $a_s$, one must avoid large logarithms at higher orders. For the term with $U(\mu, \nu)$ this requires one to take $\nu \sim \mu$. Satisfying this condition, one obtains large logarithms in (89), so that the choice $\nu \ll \Delta$ has to be avoided. On the other hand, no large logarithms appear for $\Delta \ll \nu$ according to (87). Fixed order matching is thus possible both for $\Delta \sim \nu$ and for $\Delta \ll \nu$. The latter choice is in fact unavoidable for matching at the point $\Delta = 0$.

For $j = 0, 1, 2$ a closed form of $I_j$ can be given in terms of the generalised hypergeometric functions $_pF_q$, namely

$$I_0(\Delta, \nu, \nu) = -\frac{a^2}{4} \, {}_2F_3\left(1, 1; 2, 2, 2; -\frac{a^2}{4}\right),$$

$$I_1(\Delta, \nu, \nu) = \frac{a^2}{4} \, {}_3F_4\left(1, 1, 1; 2, 2, 2, 2; -\frac{a^2}{4}\right),$$

$$I_2(\Delta, \nu, \nu) = -\frac{a^2}{2} \, {}_4F_5\left(1, 1, 1, 1; 2, 2, 2, 2, 2; -\frac{a^2}{4}\right). \tag{90}$$

## 3.3 Scale independence of matching

We now discuss the dependence of the matching equation (83) on the cut-off scale $\nu$ and its behaviour under renormalisation group evolution. Let us first verify explicitly that the $\nu$ dependence is the same on the left and right hand sides. The logarithmic derivative of the l.h.s. is

$$-\frac{d}{d\log \nu^2} F_\Phi(\Delta, \mu, \nu) = -\frac{d}{d\log \nu^2} \pi \int_{b_0^2/\nu^2}^{\infty} dy^2 \, J_0(y\Delta) F(y, \mu)$$

$$= -\frac{\pi b_0^2}{\nu^2} J_0\left(\frac{b_0 \Delta}{\nu}\right) F\left(y = \frac{b_0}{\nu}, \mu\right)$$

$$= -J_0\left(\frac{b_0 \Delta}{\nu}\right) \sum_{n=1}^{\infty} a_s^n \sum_{k=0}^{n-1}\left(\log \frac{\mu^2}{\nu^2}\right)^k V^{[n,k]} \underset{12}{\otimes} f(\mu), \tag{91}$$

where in the last step we used the perturbative splitting form of $F(y)$ in (25) and (59), as is appropriate for $y = b_0/\nu$. For the r.h.s. of (83) one gets

$$\frac{d}{d\log \nu^2} \sum_{n=1}^{\infty} a_s^n \left\{\sum_{k=0}^{n}\left(\log \frac{\mu^2}{\nu^2}\right)^k U^{[n,k]} + \sum_{k=0}^{n-1} I_k(\Delta, \nu, \mu) V^{[n,k]}\right\} \underset{12}{\otimes} f(\mu)$$

$$= -\sum_{n=1}^{\infty} a_s^n \sum_{k=0}^{n-1}\left(\log \frac{\mu^2}{\nu^2}\right)^k \left\{(k+1) U^{[n,k+1]} + \left[J_0\left(\frac{b_0 \Delta}{\nu}\right) - 1\right] V^{[n,k]}\right\} \underset{12}{\otimes} f(\mu) \tag{92}$$

using the integral representation (84) of $I_k$. With the relation (82) between $V$ and $U$, we see that both sides of the matching equation have the same $\nu$ dependence at each fixed order in $a_s$, as it should be. This statement holds up to power corrections in $\Lambda/\nu$, due to replacing $F(y)$ with its perturbative splitting form.

Let us now consider the renormalisation scale dependence of (83). Using the appropriate – homogeneous or inhomogeneous – double DGLAP equations, one obtains for the l.h.s.

$$\frac{d}{d\log\mu^2}\Big[F(\Delta,\mu)-F_\Phi(\Delta,\mu,\nu)\Big] = P_s \underset{12}{\otimes} f(\mu)$$
$$+ P \underset{1}{\otimes}\Big[F(\Delta,\mu)-F_\Phi(\Delta,\mu,\nu)\Big] + P \underset{2}{\otimes}\Big[F(\Delta,\mu)-F_\Phi(\Delta,\mu,\nu)\Big], \tag{93}$$

where the term with $P_s$ arises from the $\overline{\text{MS}}$ momentum space DPD. On the r.h.s. of (93) one can replace the difference of DPDs with the matching expression (83). Comparing this to the renormalisation group derivative of the r.h.s. of (83), one finds that the matching equation holds at all $\mu$ if

- $U(\mu,\nu) \underset{12}{\otimes} f(\mu)$ obeys the inhomogeneous double DGLAP equation. At LO and NLO, this can be explicitly verified using the evolution equation for $W(\mu,\nu)\underset{12}{\otimes} f(\mu)$ and the equality of $W$ and $U$ stated in (79),

- the $\Delta$ dependent term

$$\sum_{n=1}^\infty a_s^n(\mu) \sum_{k=0}^{n-1} I_k(\Delta,\nu,\mu) V^{[n,k]} \underset{12}{\otimes} f(\mu) \tag{94}$$

satisfies the homogeneous double DGLAP equation. This follows from the fact that the splitting form

$$\sum_{n=1}^\infty a_s^n(\mu) \sum_{k=0}^{n-1}\Big(\log\frac{y^2\mu^2}{b_0^2}\Big)^k V^{[n,k]} \underset{12}{\otimes} f(\mu) \tag{95}$$

of $F(y,\mu)$ satisfies the homogeneous equation and that $I_k(\Delta,\nu,\mu)$ and $\log^k(y^2\mu^2/b_0^2)$ satisfy the same differential equation in $\mu$.

As is the case for the perturbative splitting forms of $F(\Delta,\mu)$ and $F(y,\mu)$, the double DGLAP equations are fulfilled only up to order $a_s^n$ if the perturbative series for the splitting or matching kernel is truncated at order $a_s^n$.

## 4 Two-loop calculation

The aim of this work is to obtain the kernels $V$, $W$, $P_s$, and $U$ introduced in sections 2 and 3 at NLO accuracy. To this end it suffices to calculate only the momentum space kernel $W$, since from this $V$, $P_s$, and $U$ can readily be obtained as shown in the sections just mentioned. In this section, we describe the computation of the NLO momentum space kernel $W^{(2)}_{a_1 a_2,b}(\Delta)$ in some detail.

To begin with, we apply the factorisation formula (24) to the bare DPD $F_{B,a_1 a_2/a_0}(\Delta)$ of partons $a_1$ and $a_2$ in a parton $a_0$. Expanding the formula in $a_s$, we obtain for the second order term

$$F^{(2)}_{B,a_1 a_2/a_0}(\Delta) = \sum_b\Big[W^{(2)}_{B,a_1 a_2,b}(\Delta)\underset{12}{\otimes} f^{(0)}_{B,b/a_0} + W^{(1)}_{B,a_1 a_2,b}(\Delta)\underset{12}{\otimes} f^{(1)}_{B,b/a_0}\Big]$$
$$= W^{(2)}_{B,a_1 a_2,a_0}(\Delta), \tag{96}$$

where $f_{B,b/a_0}$ denotes the bare PDF of parton $b$ in parton $a_0$. In the second step we used the tree-level expression $f^{(0)}_{B,b/a_0}(x) = \delta_{b a_0}\delta(1-x)$ and the fact that $f^{(1)}_{B,b/a_0} = 0$ because the

corresponding loop integrals do not depend on any dimensionful scale. We thus obtain the bare two-loop kernel directly from the two-loop graphs for the bare DPD of partons $a_1$ and $a_2$ in an on-shell parton $a_0$. We compute with massless quarks and gluons, so that the relevant graphs have both infrared and ultraviolet divergences, both being treated by dimensional regularisation.

## 4.1 Channels and graphs

Let us analyse at which order in $a_s$ a kernel $W_{a_1 a_2, a_0}$ is first nonzero. The conservation of quark and antiquark flavour implies that the splitting $a_0 \to a_1 a_2$ starts at

1. LO for $g \to g g$, $g \to q \bar{q}$ and $q \to q g$,

2. NLO for $g \to q g$, $q \to g g$ and $q_j \to q_j q_k$, $q_j \to q_j \bar{q}_k$, $q_j \to q_k \bar{q}_k$,

where $j$ and $k$ label quark flavours and may be equal or different. We will refer to these as "LO channels" and "NLO channels", respectively. Further channels at the same orders are obtained by interchanging partons $a_1$ and $a_2$ or by charge conjugation, where the kernels for charge conjugated channels are identical. All other splitting processes start either at NNLO or $N^3$LO.

Real emission graphs (or "real graphs" for short) are shown in figures 1 and 2 for LO and NLO channels, respectively. Further graphs for the same channels are obtained by complex conjugation and by interchanging the lines for partons $a_1$ and $a_2$ if $a_1 = a_2$. The unobserved parton radiated into the final state is uniquely determined for given $a_0$, $a_1$, and $a_2$: it is a gluon for the LO channels and a quark or antiquark for the NLO channels.

For the LO channels, we have also virtual loop graphs (or "virtual graphs" for short). They are obtained by dressing the LO splitting graphs with one vertex correction or with one propagator correction for the lines of parton $a_1$ or $a_2$ (propagator corrections for the massless on-shell parton $a_0$ are zero in dimensional regularisation). This is depicted in figure 3.

At this point a brief comment is in order about the flavour structure for channels where $a_0, a_1$ and $a_2$ are only quarks or antiquarks. In graphs 2k, 2l and 2n there are separate fermion lines for quarks $q$ and $q'$, which may or may not have the same flavour. We denote the corresponding kernels $W_{a_1 a_2, a_0}$ as specified in the figure. By contrast, graphs 2m and 2o are of a valence type in the sense that they involve a single quark flavour; we denote the corresponding kernels with a superscript $v$. The complete kernels for a specific flavour transition are thus obtained as

$$
\begin{aligned}
W_{q_j \bar{q}_k, q_i} &= \delta_{jk} W_{q' \bar{q}', q} + \delta_{ij} W_{q \bar{q}', q} + \delta_{ij} \delta_{jk} W_{q \bar{q}, q}^v , \\
W_{q_j q_k, q_j} &= \delta_{jk} W_{q' q, q} + \phantom{\delta_{ij}} W_{q q', q} + \phantom{\delta_{ij}} \delta_{jk} W_{q q, q}^v ,
\end{aligned}
\tag{97}
$$

where $W_{q'q,q}(x_1, x_2) = W_{qq',q}(x_2, x_1)$.

**Feynman vs. light-cone gauge.** We performed two independent calculations of the graphs making up the bare partonic DPDs – one in covariant Feynman gauge, and the other in $A^+ = 0$ light-cone gauge. For cut graphs in Feynman gauge, we use the Feynman rules given in appendix D of [73]. Not given there are the expressions for the four-gluon vertex. The expression to be used to the left of the cut can readily be found in textbooks (see e.g. section 3.2.4 in [70]), whereas to the right of the cut one needs to take the complex conjugate expression (just as for the quark-gluon vertex).

In addition to the graphs shown in figures 1, 2, and 3, one gets in Feynman gauge also graphs with Wilson lines attaching to the active partons, as already mentioned below (3). These Wilson line graphs can be obtained from the graphs without Wilson lines following the prescription illustrated in figure 4. Generally speaking, Wilson lines appear in two cases. The

(a) LD  (b) UD  (c) UND  (d) T2B

(e)  (f)  (g)

(h)  (i)  (j)  (k)

(l)  (m)  (n)  (o)

(p)  (q)

Figure 1: Real graphs for LO channels. The parton lines at the bottom of the graph correspond to $a_0$, and those on top to $a_1$, $a_2$, $a_2$ and $a_1$ from left to right. The topologies "LD", "UD" etc. are explained in section 4.2.

Figure 2: Real graphs for NLO channels. The association of parton lines is as in figure 1.

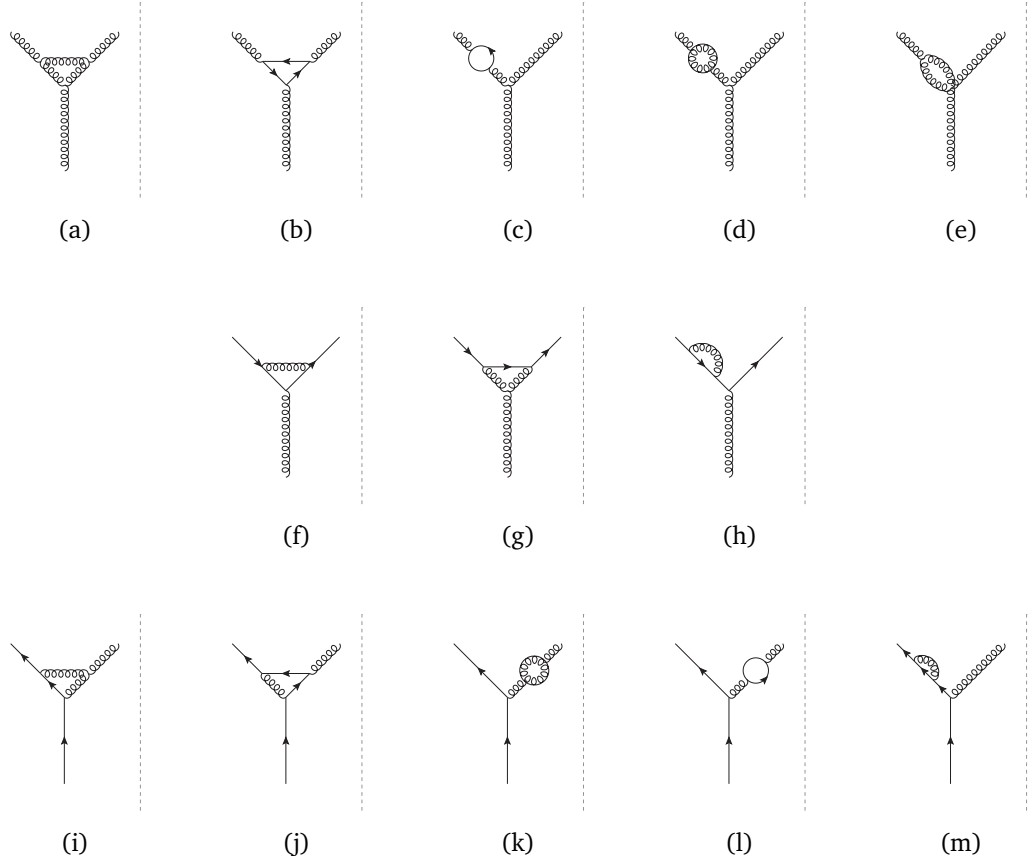

Figure 3: Virtual graphs for the amplitude in LO channels. The tree graphs in the complex conjugate amplitude are not shown.

first is when an active parton, $a_1$ or $a_2$, is a gluon emerging from a three-gluon vertex on one of the upper legs, as for instance in figures 1a, 1b, and 3a. Such graphs give rise to two kinds of Wilson line graphs as shown in figure 4a and 4b. In the second case, the active parton is a quark or antiquark originating from a quark-gluon vertex on one of the upper legs, as is e.g. the case in figures 1h, 1l, 3f, and 3g. This leads to only one Wilson line graph with a gluon attaching to a quark Wilson line, visualised in figure 4c and 4d.

Finally, in Feynman gauge one has to include the corresponding Fadeev-Popov ghost version for all graphs containing closed gluon loops, such as in figure 3a and 3d.

**Kinematics.**    For real graphs, both in LO and NLO channels, the general kinematic structure is illustrated in figure 5. We work in a frame where the incoming parton $a_0$ has plus momentum $p^+$ and zero transverse and minus momentum. The plus momenta $x_1 p^+$ and $x_2 p^+$ of the active partons $a_1$ and $a_2$ are equal in the amplitude and its conjugate, whereas their transverse momenta differ by $\pm\boldsymbol{\Delta}$ as shown in figure 5. Minus momenta are assigned in the same way as the transverse ones. The parton $a_3$ that goes into the final state has plus momentum $x_3 p^+$, and its momentum is uniquely fixed by the momenta of the active partons $a_1$ and $a_2$, i.e.

$$x_3 = 1 - x_1 - x_2, \qquad k_3^- = -k_1^- - k_2^-, \qquad \boldsymbol{k}_3 = -\boldsymbol{k}_1 - \boldsymbol{k}_2. \qquad (98)$$

To obtain collinear momentum space DPDs, we have to integrate the expressions corresponding to the real graphs in figures 1 and 2 over $k_1^-$, $k_2^-$, and $\Delta^-$, which corresponds to having $z_1^+ = z_2^+ = y^+ = 0$ in the matrix element (1). As we are interested in collinear DPDs, we

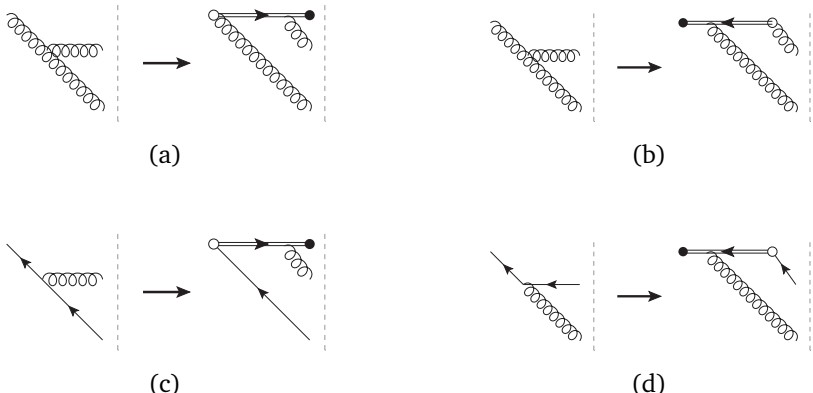

Figure 4: Rules for obtaining the Wilson line graphs needed in Feynman gauge from graphs without Wilson lines. In each of the four panels, the top left parton is an active one ($a_1$ or $a_2$). These rules apply to both real and virtual graphs, and corresponding rules hold on the right of the final state cut. Note that the three-gluon vertex in the upper panels gives rise to two different graphs.

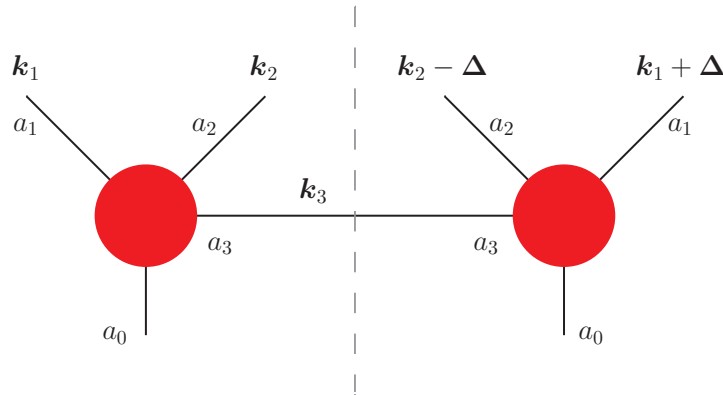

Figure 5: Assignment of transverse momenta in of a real graph. Note that compared with the symmetric assignment in [43] we have shifted the integration variables $\boldsymbol{k}_1$ and $\boldsymbol{k}_2$ such that $\boldsymbol{\Delta}$ appears only on the r.h.s. of the cut.

also have to integrate over the transverse momenta $\boldsymbol{k}_1$ and $\boldsymbol{k}_2$, which is tantamount to the condition $\boldsymbol{z}_1 = \boldsymbol{z}_2 = \boldsymbol{0}$ in (1).

The kinematic structure of the virtual graphs we encounter in the LO channels is exemplified in figure 6. Due to momentum conservation we now find that $k_2^- = -k_1^-$ and $\boldsymbol{k}_2 = -\boldsymbol{k}_1$, so that the integrations over $k_2^-$ and $\boldsymbol{k}_2$ become redundant. However, there now is an additional loop momentum $\ell$ over which we have to integrate.

## 4.2 Performing the calculation

We now move on to outline the details of our calculation in Feynman and light-cone gauge. The initial computation of the diagram expressions is done using FORM [74] or FeynCalc [75].[4] One then needs to perform the integrations over phase space and loop momenta. As mentioned before, there are two types of graph contributing to the bare partonic DPD at NLO:

---

[4]In practice, FORM was used in the light-cone gauge calculation, whilst FeynCalc was used for the Feynman gauge calculation. Of course, either code could have been used for both computations.

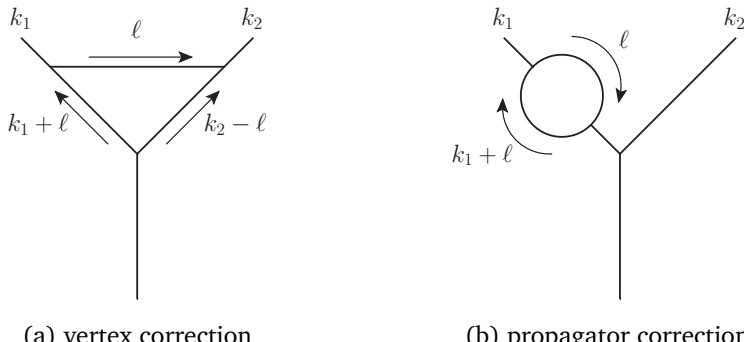

(a) vertex correction  (b) propagator correction

Figure 6: Assignment of momenta in generic virtual graphs to the left of the cut. An analogous assignment holds for loops to the right of the cut, with $k_1$ and $k_2$ shifted by $+\Delta$ and $-\Delta$, respectively.

real emission graphs in which one parton is emitted into the final state, and virtual loop graphs with a loop in the amplitude or its conjugate and no emitted particles in the final state. We discuss the methodology for performing the integrations for the two cases in turn.

### 4.2.1 Real emission diagrams

For the real emission diagrams we have four different topologies. These are exemplified by figure 1a, 1b, 1c, 1d, and we refer to them as LD (lower diagonal), UD (upper diagonal), UND (upper non-diagonal) and T2B (top to bottom). The four-gluon graphs in figure 1e, 1f, and 1g can be considered to fit into the T2B topology, because all the denominator factors in these diagrams are covered by those in the T2B topology.

For each graph, we begin by performing the integration over minus components of the loop and phase space momenta. This is most easily done by changing variables as follows:

$$K^- = k_1^- + k_2^-, \qquad k^- = (k_1^- - k_2^-)/2, \qquad k'^- = (k_1^- - k_2^-)/2 + \Delta^-. \qquad (99)$$

The integral over $K^-$ is performed using the on-shell delta function, whilst the integrals over $k^-$ and $k'^-$ can be performed using complex contour integration. For all topologies, one is able to close the integration contour on only one pole, both in the $k^-$ and in the $k'^-$ integration. We note in passing that the variable transformation (99) is also useful in the general proofs of [54, 76].

As all plus components are fixed by the operators in (1), it remains to integrate over $\boldsymbol{k}_1$ and $\boldsymbol{k}_2$ at fixed $\boldsymbol{\Delta}$. Every graph may be written in a form where its denominator factors are contained within the following set (this may require some relabelling of integration momenta, and use of the fact that the integrals are invariant under $\boldsymbol{\Delta} \to -\boldsymbol{\Delta}$):

$$D_1 = \frac{(\boldsymbol{k}_1 + \boldsymbol{\Delta})^2}{x_1} + \frac{(\boldsymbol{k}_2 - \boldsymbol{\Delta})^2}{x_2} + \frac{(\boldsymbol{k}_1 + \boldsymbol{k}_2)^2}{x_3}, \qquad D_2 = \frac{\boldsymbol{k}_1^2}{x_1} + \frac{\boldsymbol{k}_2^2}{x_2} + \frac{(\boldsymbol{k}_1 + \boldsymbol{k}_2)^2}{x_3},$$

$$D_3 = (\boldsymbol{k}_1 + \boldsymbol{\Delta})^2, \qquad\qquad D_4 = \boldsymbol{k}_2^2,$$

$$\tilde{D}_4 = \boldsymbol{k}_1^2, \qquad\qquad \tilde{D}_5 = (\boldsymbol{k}_1 + \boldsymbol{k}_2)^2. \qquad (100)$$

Graphs of all topologies have denominators $D_1$ and $D_2$. In addition LD graphs have $\tilde{D}_5$, UD graphs have $D_3$ and $\tilde{D}_4$, T2B graphs have $D_3$ and $\tilde{D}_5$, and UND graphs have $D_3$ and $D_4$.

We express all numerator factors in the graphs in terms of the $D_i$ and $\tilde{D}_i$, and then use integration by parts identities [77, 78], as implemented in LiteRed [79], to express the results

for all graphs in terms of a small set of master integrals. For graphs with the topology UND the master integrals have the general form

$$I_1(\alpha_1, \alpha_2, \alpha_3, \alpha_4) = \int \frac{d^{d-2}\boldsymbol{k}_1 \, d^{d-2}\boldsymbol{k}_2}{\prod_{i=1..4} D_i^{\alpha_i}} \,. \tag{101}$$

Specifically, we need the integrals

$$\begin{aligned} I_1(1,1,0,0), \quad &I_1(0,1,1,0), \quad I_1(1,1,1,0), \\ I_1(1,0,1,1), \quad &I_1(1,1,1,1), \quad I_1(2,1,1,1), \end{aligned} \tag{102}$$

as well as integrals related to these by the simple transformation $x_1 \leftrightarrow x_2$.

For the remaining topologies, the master integrals are of the form

$$I_2(\alpha_1, \alpha_2, \alpha_3, \alpha_4, \alpha_5) = \int \frac{d^{d-2}\boldsymbol{k}_1 \, d^{d-2}\boldsymbol{k}_2}{\prod_{i=1...3} D_i^{\alpha_i} \, \prod_{i=4...5} \tilde{D}_i^{\alpha_i}} \,, \tag{103}$$

and specifically we need

$$I_2(0,1,1,0,1), \quad I_2(1,1,1,1,0), \tag{104}$$

as well as the integrals on the first line of (102) and integrals related to these by $x_1 \leftrightarrow x_2$.

We apply the method of differential equations [80–83] to compute these master integrals. The integrals depend on the external variables $x_1$, $x_2$ and $\Delta^2$. Since $\Delta^2$ is the only dimensionful external quantity, each master integral only depends on $\Delta^2$ via an overall prefactor of a power of $\Delta^2$ that is easily determined by dimensional analysis. The differential equations in $\Delta^2$ thus only provide trivial information and are not used. We compute the master integrals by solving the differential equations in $x_1$. For a particular master integral, solving the equation determines the integral up to a "constant" of integration, which here is actually a function of $x_2$. We determine this constant of integration by computing the leading behaviour of the master integrals in the vicinity of the line $x_3 = 0$. As a cross-check, we then verify that the obtained solution satisfies the differential equation in $x_2$.

The system of differential equations has an almost entirely triangular matrix structure. For the $I_1$ integrals the system schematically looks as follows:

$$\frac{\partial}{\partial x_1} \begin{bmatrix} I_1(1,1,0,0) \\ I_1(0,1,1,0) \\ I_1(1,1,1,0) \\ I_1(1,0,1,1) \\ I_1(1,1,1,1) \\ I_1(2,1,1,1) \end{bmatrix} = \begin{bmatrix} \blacksquare & 0 & 0 & 0 & 0 & 0 \\ 0 & \blacksquare & 0 & 0 & 0 & 0 \\ \blacklozenge & \blacklozenge & \blacksquare & 0 & 0 & 0 \\ 0 & \blacklozenge & 0 & \blacksquare & 0 & 0 \\ \blacklozenge & \blacklozenge & \blacklozenge & \blacklozenge & \blacksquare & \blacksquare \\ \blacklozenge & \blacklozenge & \blacklozenge & \blacklozenge & \blacksquare & \blacksquare \end{bmatrix} \begin{bmatrix} I_1(1,1,0,0) \\ I_1(0,1,1,0) \\ I_1(1,1,1,0) \\ I_1(1,0,1,1) \\ I_1(1,1,1,1) \\ I_1(2,1,1,1) \end{bmatrix} \,, \tag{105}$$

where black squares $\blacksquare$ denote entries of the form $c(x_1, x_2)$ while black diamonds $\blacklozenge$ denote entries of the form $c_1(x_1, x_2) + c_2(x_1, x_2) P_{x_1 \leftrightarrow x_2}$. Here $P_{x_1 \leftrightarrow x_2}$ is the operator transforming a given master integral $I_1(\alpha_1, \alpha_2, \alpha_3, \alpha_4)$ to the corresponding master integral with $x_1 \leftrightarrow x_2$. This enables us to solve most of the system using elementary methods for first-order differential equations and forward substitution, working downwards from $I_1(1,1,0,0)$ and $I_1(0,1,1,0)$. The two-denominator integrals can be solved exactly in $d = 4 - 2\epsilon$ dimensions, whilst for the remainder we obtain a series in $\epsilon$ up to the required order. The only instance in which we have a coupled system of differential equations is for $I_1(1,1,1,1)$ and $I_1(2,1,1,1)$. Here we transform the $2 \times 2$ system to the canonical basis [84] using Fuchsia [85], which allows us to obtain a solution for these two integrals as a series in $\epsilon$. The system of differential equations

for the $I_2$ integrals is fully triangular, so that application of elementary methods plus forward substitution suffices in this case.

The leading behaviour of the integrals for $x_3 \to 0$ is computed using the method of regions [86]. Let us introduce the scaling parameter $\lambda \ll 1$, and say that $x_3 \sim \lambda$. Then for all master integrals the following region gives a leading contribution in $\lambda$:

$$R_1: \quad x_3 \sim \lambda, \quad x_1, x_2 \sim 1, \quad \boldsymbol{k}_1^2, \boldsymbol{k}_2^2, (\boldsymbol{k}_1 + \boldsymbol{\Delta})^2 \sim \Delta^2, \quad (\boldsymbol{k}_1 + \boldsymbol{k}_2)^2 \sim \lambda \Delta^2. \quad (106)$$

For $I_1(1,0,1,1)$ only, we identify a further leading region, namely

$$R_2: \quad x_3 \sim \lambda, \quad x_1, x_2 \sim 1, \quad \boldsymbol{k}_1^2, \boldsymbol{k}_2^2, (\boldsymbol{k}_1 + \boldsymbol{\Delta})^2, (\boldsymbol{k}_1 + \boldsymbol{k}_2)^2 \sim \Delta^2. \quad (107)$$

For each region, we use the appropriate scaling and drop terms in the denominator that are subleading in $\lambda$ (so that the result is homogeneous in $\lambda$). Following this approximation, every master integral has a sufficiently simple form to be solved to all orders in $\epsilon$ by the method of Feynman parameters. Then one adds together the contributions from the leading regions to obtain the leading behaviour in the limit $x_3 \to 0$. For any master integral, the region $R_1$ gives a non-integer power of $x_3$ for $\epsilon \neq 0$, namely $x_3^{1-\epsilon}$. This is because all denominators behave like $\lambda^0$, whilst the phase space contributes $\lambda^{1-\epsilon}$. For $I_1(1,0,1,1)$ the region $R_2$ gives an integer power of $x_3$, namely $x_3^1$, because $D_1$ behaves like $\lambda^{-1}$, whilst all other denominators and the phase space behave like $\lambda^0$.

For each master integral, we checked our analytic results against a numerical computation of the integral performed by FIESTA 2 [87] at 10 randomly chosen $(x_1, x_2)$ points. The results were found to agree within the precision of the numerical computation.

When computing the expressions for the graphs, careful consideration of the limit $x_3 \to 0$ is needed, because some graphs are singular in this limit. Fortunately, we know the full (all-order in $\epsilon$) behaviour of the graphs for $x_3 \to 0$ from the boundary condition computation for the master integrals. For graphs associated with the non-UND topologies, this singularity is always regulated by the dimensional regularisation parameter $\epsilon$, since the non-UND master integrals only have the leading region $R_1$ for $x_3 \to 0$. We make a distributional expansion of the relevant $x_3^{-1-n\epsilon}$ factors as follows:

$$\theta(x_3) x_3^{-1-n\epsilon} = -\frac{1}{n\epsilon} \delta(x_3) + \mathcal{L}_0(x_3) - n\epsilon \mathcal{L}_1(x_3) + \frac{n^2 \epsilon^2}{2} \mathcal{L}_2(x_3) + \mathcal{O}(\epsilon^3), \quad (108)$$

where we denote the plus distributions as

$$\mathcal{L}_n(x) = \left[ \frac{\theta(x) \ln^n(x)}{x} \right]_+. \quad (109)$$

Useful relations for these distributions are given in [88]. Our final results turn out to involve only $\mathcal{L}_0$ and not $\mathcal{L}_n$ with $n > 0$.

For graphs associated with the UND topologies, specifically in the contribution from $I_1(1,0,1,1)$, one can have terms which just go as $1/x_3$ and are not regulated by $\epsilon$. This corresponds to a rapidity divergence and requires an additional regulator. In covariant gauge, the rapidity regulator is typically inserted via some modification of the Wilson lines, whilst in light-cone gauge it corresponds to a modification of the gluon propagator. As noted in section 3.5 of [11] or in [64], DPDs in the colour singlet channel do not suffer from rapidity divergences.[5] Therefore, the divergences associated with $I_1(1,0,1,1)$ must cancel when we sum over graphs, including the virtual loop graphs.

---

[5]This is analogous to what happens in the single scattering sector, where for the collinear PDFs we have no rapidity divergences [89].

There are various possible choices for the rapidity regulator, such as the analytic regulator of [90], the exponential regulator of [91], the "pure rapidity regulator" of [92], the $\delta$ regulator of [93], the CMU $\eta$ regulator of [94, 95], or the use of Wilson lines tilted away from the light cone as described in [70]. We tried each of the latter three options, and we find that when consistently implementing the regulator in all real emission and virtual loop graphs, the sum over graphs yields the same rapidity-divergence-free result for all regulators. Note that we always remove the rapidity regulator before performing ultraviolet renormalisation and setting $\epsilon \to 0$, as is prescribed when using such regulators. Note also that for each regulator we have a distributional expansion analogous to (108). We will give more detail in a forthcoming publication devoted to two-loop results for the colour non-singlet channels. In this case, the rapidity divergences only cancel when one combines the bare DPDs with the appropriate soft factor.

### 4.2.2 Virtual loop diagrams

Just as for the real graphs, the integrations over minus momenta (including $\ell^-$) in virtual graphs are easily done using complex contour integration. The integrations over the transverse momenta $\boldsymbol{k}_1$ and $\boldsymbol{\ell}$ were performed using different methods in the Feynman gauge and light-cone gauge calculations. In the former case we again used integration by parts reduction, as we did for the real emission graphs. One can even re-use the same master integrals, namely $I_1(0, 1, 1, 0)$ and $I_1(0, 1, 1, 1)$, where the latter is obtained from $I_1(1, 0, 1, 1)$ by interchanging $x_1 \leftrightarrow x_2$. In the calculation using light-cone gauge, the adopted strategy closely followed that of [52] – indeed we were able to directly re-use many of the results given in Appendix A of that work.

Finally, we have to integrate $\ell^+$ over a finite range. This finite range is a result of the earlier integrations over minus components: when $\ell^+$ is outside a certain range, all poles move into one half-plane for one of these integrations, and the result is zero. The $\ell^+$ integration may be performed using standard integration techniques. However, this again requires a consistent treatment of endpoint singularities. We have to perform a distributional expansion analogous to (108) in cases where these singularities are regulated by $\epsilon$, and when this is not the case we have to insert a rapidity regulator. In light-cone gauge, the graphs containing such a rapidity divergence are vertex correction graphs in which the line with momentum $\ell$ is a gluon, such as figure 3f. In Feynman gauge the rapidity divergence is contained in the corresponding Wilson line graphs. Here we insert the same regulator as used in the real emission calculations.

## 5 Results

In this section, we present our results for the kernels $P_s$ and $W$ up to NLO and discuss some of their properties. Results for the kernels $V$ and $U$ can readily be obtained from the relations (64) and (79).

**Support and singularity structure** Let us briefly discuss the singularity structure of the NLO results. This differs between the LO and NLO channels defined at the beginning of section 4.1. The kernels for LO channels have the general form

$$K(x_1, x_2) = K_{\text{reg}}(x_1, x_2) + \frac{K_p(x_1, x_2)}{[1 - x_1 - x_2]_+} + K_\delta(x_1)\, \delta(1 - x_1 - x_2), \tag{110}$$

where $K$ is either $W^{(2)}$ or $P_s^{(1)}$. The kernels for NLO channels contain only a term $K_{\text{reg}}$. The functions $K_{\text{reg}}$ and $K_p$ are regular in the region defined by the conditions $0 < x_1$, $0 < x_2$ and

$x_1 + x_2 \leq 1$, and $K_\delta$ is regular in the region $0 < x_1 < 1$. Note that these regions exclude the points $x_1 = 1$ and $x_2 = 1$, where some kernels have power-law divergences of the form $(1 - x_1)^{-n_1}(1 - x_2)^{-n_2}$ with integers $n_1$ and $n_2$. These points are not reached in any physical double parton scattering process, where both $x_1$ and $x_2$ must be strictly positive. We note that the decomposition (110) is invariant under the simultaneous replacement

$$
\begin{aligned}
K_{\text{reg}}(x_1, x_2) &\to K_{\text{reg}}(x_1, x_2) + \varphi(x_1, x_2), \\
K_p(x_1, x_2) &\to K_p(x_1, x_2) - (1 - x_1 - x_2)\, \varphi(x_1, x_2),
\end{aligned}
\tag{111}
$$

where $\varphi(x_1, x_2)$ is a regular function. This leaves a freedom of choice in the form of $K_{\text{reg}}$ and $K_p$ for a given kernel.

Inserting the form (110) into the convolution with a PDF and using the representation (13), one obtains one-variable distributions $1/[1 - z]_+$ and $\delta(1 - z)$, so that

$$
\begin{aligned}
\left[ K \underset{12}{\otimes} f \right](x_1, x_2) &= \frac{1}{x_1 + x_2} \int_{x_1 + x_2}^{1} dz \left\{ K_{\text{reg}}(zu, z\bar{u}) + \frac{K_p(zu, z\bar{u})}{[1 - z]_+} \right\} f\left( \frac{x_1 + x_2}{z} \right) \\
&\quad + K_\delta(u) \frac{f(x_1 + x_2)}{x_1 + x_2},
\end{aligned}
\tag{112}
$$

with

$$
u = \frac{x_1}{x_1 + x_2}, \qquad\qquad \bar{u} = 1 - u. \tag{113}
$$

The term with $K_\delta$ in (112) has the same structure as the convolution of the LO kernels $W^{(1)}$ or $P_s^{(0)}$ with $f$.

Before presenting the kernels in detail, we note that

$$
\begin{aligned}
W_{q\bar{q},g}(x_1, x_2) &= W_{\bar{q}q,g}(x_1, x_2), & W_{gq,g}(x_1, x_2) &= W_{g\bar{q},g}(x_1, x_2), \\
W_{q'\bar{q}',q}(x_1, x_2) &= W_{\bar{q}'q',q}(x_1, x_2), & W_{qq',q}(x_1, x_2) &= W_{q\bar{q}',q}(x_1, x_2).
\end{aligned}
\tag{114}
$$

Analogous relations hold for the kernels $P_{a_1 a_2, a_0}$. The relations in the first line follows from charge conjugation, whereas those in the second line are obtained by reversing the quark line associated with $q'$ (but not the one associated with $q$). The channels on the r.h.s. of (114) are not discussed further in the following. We also recall that kernels for specific flavour transitions are obtained from (97) and its analogue for $P_{a_1 a_2, a_0}$, and that kernels for channels with an initial $\bar{q}$ are equal to the kernels for the charge conjugate channels with an initial $q$.

Useful building blocks for presenting the explicit kernels are the functions

$$
\begin{aligned}
p_{gg}(x) &= \frac{x}{1 - x} + \frac{1 - x}{x} + x(1 - x), & p_{qq}(x) &= \frac{1 + x^2}{1 - x}, \\
p_{qg}(x) &= x^2 + (1 - x)^2, & p_{gq}(x) &= \frac{1 + (1 - x)^2}{x},
\end{aligned}
\tag{115}
$$

which are proportional to the LO DGLAP evolution kernels away from the endpoint $x = 1$. The LO kernels $P_s^{(0)}$ and $W^{[1,0]}$ read

$$
\begin{aligned}
P_{gg,g}^{(0)}(x) &= 2C_A\, p_{gg}(x) & W_{gg,g}^{[1,0]}(x) &= 0, \\
P_{q\bar{q},g}^{(0)}(x) &= T_F\, p_{qg}(x), & W_{q\bar{q},g}^{[1,0]}(x) &= -2T_F\, x(1 - x) \\
P_{qg,q}^{(0)}(x) &= C_F\, p_{qq}(x), & W_{qg,q}^{[1,0]}(x) &= -C_F\,(1 - x).
\end{aligned}
\tag{116}
$$

We find it useful to express the NLO kernels in terms of the overcomplete set of momentum fractions $x_1, x_2$ and $x_3 = 1 - x_1 - x_2$. This makes symmetry properties more transparent and often allows for more compact expressions. Furthermore we will use the notation $\bar{x}_i = 1 - x_i$. Let us first discuss the evolution kernels $P_s^{(1)}$.

$1 \to 2$ **evolution kernels.**  Starting with the pure gluon channel, we introduce auxiliary functions

$$
R_{gg,g}(x_1, x_2, x_3) = C_A^2 \left\{ \frac{\log(1 + x_1/x_3) + \log(1 + x_1/x_2)}{x_1} \frac{x_2 \, p_{gg}(x_2)}{\bar{x}_3} \right.
$$

$$
+ 2 \left[ \frac{x_2^2(3 + 4x_2)}{\bar{x}_1^4} - \frac{x_2(4 + 9x_2 + x_2^2)}{\bar{x}_1^3} + \frac{1 + 6x_2 + 2x_2^2}{\bar{x}_1^2} - \frac{1 + x_2}{\bar{x}_1} + \frac{1}{2} \right]
$$

$$
+ \log x_1 \left[ \frac{4x_1^2}{\bar{x}_3^4} - \frac{4x_1(1 + x_1)}{\bar{x}_3^3} + \frac{4(2 + x_1 + 2x_1^2)}{\bar{x}_3^2} - \frac{3 - 3x_1 + x_1^2 + x_1^3}{\bar{x}_1^4} x_2^2 \right.
$$

$$
\left. + \frac{10 - 25x_1 + 12x_1^2 - 3x_1^3}{2\bar{x}_1^2} - \frac{2 + 7x_1 - 3x_1^2}{\bar{x}_2} - \frac{2}{\bar{x}_1 \bar{x}_2 \bar{x}_3} \right]
$$

$$
- \frac{\log \bar{x}_1}{x_1} \left[ \frac{2 + 3x_1 - 9x_1^2 + 9x_1^3 - x_1^4}{\bar{x}_1^4} x_2^2 + \frac{6 + 4x_1 - 5x_1^2 + 10x_1^3 - 3x_1^4}{2\bar{x}_1^2} \right.
$$

$$
\left. \left. - \frac{4 - 6x_1 + 9x_1^2 - 6x_1^3 + 3x_1^4}{\bar{x}_1 \bar{x}_2} \right] \right\},
$$

$$
D_{gg,g}(x_1) = - C_A^2 \left\{ \frac{3}{4} + 2 p_{gg}(x_1) \left[ \log x_1 \log \bar{x}_1 + \frac{\pi^2}{6} - \frac{2}{3} \right] \right\}
$$

$$
+ \beta_0 \, C_A \, \frac{40 - 27x_1 - 20x_1^3}{12x_1}, \tag{117}
$$

in terms of which the $1 \to 2$ evolution kernel can be written as

$$
P_{gg,g}^{(1)}(x_1, x_2, x_3) = R(x_1, x_2, x_3) + R(x_2, x_1, x_3) + R(x_3, x_2, x_1)
$$

$$
+ R(x_1, x_3, x_2) + R(x_3, x_1, x_2) + R(x_2, x_3, x_1) + \left[ D(x_1) + D(x_2) \right] \delta(x_3). \tag{118}
$$

Here we dropped the subscripts on $R$ and $D$ for brevity. If we replace $x_3 = 1 - x_1 - x_2$ and express $R$ as a function of only $x_1$ and $x_2$, we have

$$
P_{gg,g}^{(1)}(x_1, x_2) = R(x_1, x_2) + R(x_2, x_1) + R(1 - x_1 - x_2, x_2) + R(x_1, 1 - x_1 - x_2)
$$

$$
+ R(1 - x_1 - x_2, x_1) + R(x_2, 1 - x_1 - x_2) + \left[ D(x_1) + D(x_2) \right] \delta(1 - x_1 - x_2). \tag{119}
$$

We see in (118) that, apart from the distribution term involving $\delta(x_3)$, the kernel is fully symmetric in the momentum fractions of the three final-state gluons. The symmetry under the interchange $x_1 \leftrightarrow x_2$ is trivial, but the symmetry between an observed gluon ($a_1$ or $a_2$) and the unobserved one ($a_3$) is not. We discuss this "active-spectator symmetry" (where we refer to $a_1, a_2$ as active partons and to $a_3$ as the spectator) in section 5.2.

For the $g \to q\bar{q}$ channel, we define

$$
\begin{aligned}
R_{q\bar{q},g}(x_1, x_2, x_3) = C_A T_F &\left\{ \frac{\log(1 + x_3/x_2)}{x_3} \frac{p_{qg}(x_1) + p_{qg}(x_2)}{2} \right. \\
&- x_1 x_2 \frac{3 - 10 x_3 + 3 x_3^2}{\bar{x}_3^4} + \frac{1 - 6 x_3 + 5 x_3^2 - 2 x_3^3}{2 \bar{x}_3^2} \\
&- 2 \log x_1 \left[ \frac{2 x_1^2}{\bar{x}_3^4} - \frac{2 x_1(1 + x_1)}{\bar{x}_3^3} + \frac{1 + 2 x_1 + 4 x_1^2}{\bar{x}_3^2} - \frac{1 + 6 x_1 - 2 x_1^2}{2 \bar{x}_3} + 1 - x_1 - \frac{x_3}{2} \right] \\
&+ 2 \log \bar{x}_1 \left[ \frac{1 - 2 x_1 + 2 x_1^2}{2 \bar{x}_3} + 1 - x_1 - \frac{x_3}{2} \right] \\
&- 2 \log x_3 \left[ x_1 x_2 \frac{2 - 3 x_3 + 2 x_3^2}{\bar{x}_3^4} - \frac{1 - x_3 + x_3^2}{2 \bar{x}_3^2} \right] \\
&\left. - \frac{2 \log \bar{x}_3}{x_3} \left[ x_1 x_2 \frac{1 + x_3 - 3 x_3^2 + 3 x_3^3}{\bar{x}_3^4} - \frac{1 + x_3 - 2 x_3^2 + 2 x_3^3}{2 \bar{x}_3^2} \right] \right\} \\
+ C_F T_F &\left\{ - \left[ \frac{3 x_3}{\bar{x}_1^2} - \frac{1 + 5 x_3}{\bar{x}_1} + 3 + 2 x_3 \right] + 2 \log x_1 \left[ \frac{x_3}{2 \bar{x}_1^2} - \frac{1 + x_3}{\bar{x}_1} - \frac{x_3}{2 \bar{x}_2^2} + \frac{1 + x_3}{\bar{x}_2} \right] \right. \\
&\left. + 2 (2 \log \bar{x}_1 - \log x_3) \left[ \frac{x_3}{2 \bar{x}_1^2} - \frac{1 + x_3}{\bar{x}_1} + 1 \right] \right\},
\end{aligned}
$$

$$
S_{q\bar{q},g}(x_1) = -2(C_A - 2 C_F) T_F x_1 \bar{x}_1,
$$

$$
\begin{aligned}
D_{q\bar{q},g}(x_1) = C_A T_F &\left\{ x_1^2 - \log^2 x_1 \, p_{qq}(x_1) \right\} + C_F T_F \left\{ \frac{3}{2} - 2 x_1 \bar{x}_1 \big( \log x_1 + \log \bar{x}_1 \big) \right. \\
&\left. + \left[ \log^2 x_1 - \log x_1 \log \bar{x}_1 - \frac{\pi^2}{6} + \frac{3}{2} \right] p_{qg}(x_1) \right\},
\end{aligned} \tag{120}
$$

and have an evolution kernel

$$
\begin{aligned}
P_{q\bar{q},g}^{(1)}(x_1, x_2, x_3) = R_{q\bar{q},g}(x_1, x_2, x_3) + R_{q\bar{q},g}(x_2, x_1, x_3) + \frac{S_{q\bar{q},g}(x_1) + S_{q\bar{q},g}(x_2)}{[x_3]_+} \\
+ \left[ D_{q\bar{q},g}(x_1) + D_{q\bar{q},g}(x_2) \right] \delta(x_3).
\end{aligned} \tag{121}
$$

The evolution kernel for the channel $g \to qg$ can be expressed by the same functions as

$$
P_{qg,g}^{(1)}(x_1, x_2, x_3) = R_{q\bar{q},g}(x_1, x_3, x_2) + R_{q\bar{q},g}(x_3, x_1, x_2) + \frac{S_{q\bar{q},g}(x_1) + S_{q\bar{q},g}(x_3)}{x_2}. \tag{122}
$$

As we did in the pure gluon channel, we observe active-spectator symmetry for the splitting process $g \to q\bar{q}g$. We obtain the kernel $P_{q\bar{g},g}^{(1)}(x_1, x_2, x_3)$ from $P_{q\bar{q},g}^{(1)}(x_1, x_2, x_3)$ if we drop the distributions terms, omitting $\delta(x_3)$ and replacing $1/[x_3]_+$ with $1/x_3$, and then interchange $x_2 \leftrightarrow x_3$.

Turning to the channel $q \to qg$, we define

$$R_{qg,q}(x_1, x_2, x_3) = C_A C_F \left\{ -\frac{\log(1 + x_3/x_1) + \log(1 - x_3/\bar{x}_1)}{x_3} \frac{x_2 p_{gq}(x_2)}{\bar{x}_1} \right.$$

$$+ 2\left[\frac{3x_1}{\bar{x}_1^4} x_3^2 - \frac{x_1}{\bar{x}_1^2} - \frac{3}{4}\right] - \left(\log x_1 + 2\log\bar{x}_1\right)\left[x_1\frac{2x_2^2}{\bar{x}_1^4} + \frac{3 + x_2^2}{\bar{x}_1^2} - \frac{1}{\bar{x}_1}\right]$$

$$\left. + 2\log x_2\left[\frac{2x_2^2}{\bar{x}_1^4} - \frac{2x_2(1 + x_2)}{\bar{x}_1^3} + \frac{4 + 2x_2 + x_2^2}{\bar{x}_1^2} - \frac{2 + x_2}{\bar{x}_1} + \frac{1}{2}\right]\right\}$$

$$+ C_F^2\left\{\frac{2\log(1 + x_3/x_1)}{x_3}\frac{x_2 p_{gq}(x_2)}{\bar{x}_1} - 2\left[\frac{x_3}{\bar{x}_2^2} + \frac{1 + x_3}{2\bar{x}_2} - 1\right]\right.$$

$$- \log x_1\left[\frac{x_2}{\bar{x}_3^2} - \frac{2 - x_2}{\bar{x}_3} - \frac{4}{\bar{x}_1} + 1\right] - \log x_2\left[\frac{x_2}{\bar{x}_3^2} - \frac{2 - x_2}{\bar{x}_3} - \frac{2 - x_2}{\bar{x}_2^2}x_3 + \frac{2}{\bar{x}_2}\right]$$

$$\left. + \frac{2\log\bar{x}_2}{x_2}\left[\frac{x_3}{\bar{x}_2^2} - \frac{2}{\bar{x}_2} + \frac{x_2 p_{gq}(x_2)}{\bar{x}_1}\right]\right\},$$

$$S_{qg,q}(x_1) = 2 C_A C_F \bar{x}_1,$$

$$D_{qg,q}(x_1) = C_A C_F\left\{1 - 2\bar{x}_1\log\bar{x}_1 + \left[\log^2 x_1 - 2\log x_1\log\bar{x}_1 - 2\operatorname{Li}_2(x_1) + \frac{4}{3}\right]p_{qq}(x_1)\right\}$$

$$- C_F^2\left\{1 + 2\bar{x}_1\log x_1 + \left[\log^2 x_1 + 3\log x_1 - 2\operatorname{Li}_2(x_1) + \frac{\pi^2}{3}\right]p_{qq}(x_1)\right\}$$

$$+ \beta_0 C_F\left\{\bar{x}_1 + \left[\log x_1 + \frac{5}{3}\right]p_{qq}(x_1)\right\}, \tag{123}$$

and have an evolution kernel

$$P_{qg,q}^{(1)}(x_1, x_2, x_3) = R_{qg,q}(x_1, x_2, x_3) + R_{qg,q}(x_1, x_3, x_2) + \frac{S_{qg,q}(x_1)}{[x_3]_+} + \frac{S_{qg,q}(x_1)}{x_2}$$

$$+ D_{qg,q}(x_1)\delta(x_3). \tag{124}$$

In terms of the same functions, the evolution kernel for $q \to gg$ reads

$$P_{gg,q}^{(1)}(x_1, x_2, x_3) = R_{qg,q}(x_3, x_2, x_1) + R_{qg,q}(x_3, x_1, x_2) + \frac{S_{qg,q}(x_3)}{x_1} + \frac{S_{qg,q}(x_3)}{x_2}. \tag{125}$$

This is symmetric in the arguments $x_1$ and $x_2$, as it must be. Again we observe active-spectator symmetry, this time under the exchange $x_1 \leftrightarrow x_3$.

For quark-antiquark transitions, we have

$$P_{q'\bar{q}',q}^{(1)}(x_1, x_2, x_3) = R_{q'\bar{q}',q}(x_1, x_2, x_3) + R_{q'\bar{q}',q}(x_2, x_1, x_3), \tag{126}$$

with

$$R_{q'\bar{q}',q} = -C_F T_F\left\{x_1 x_2\frac{1 - 6x_3 + x_3^2}{\bar{x}_3^4} + \frac{x_3}{\bar{x}_3^2}\right.$$

$$\left. + \left(2\log x_1 - \log x_3 - 2\log\bar{x}_3\right)\frac{x_1^2 + x_2^2}{2\bar{x}_3^3}p_{qq}(x_3)\right\}, \tag{127}$$

and for the quark valence kernels we have

$$P_{qq,q}^{v(1)}(x_1, x_2, x_3) = R_{qq,q}^v(x_1, x_2, x_3) + R_{qq,q}^v(x_2, x_1, x_3), \tag{128}$$

with

$$R^v_{qq,q}(x_1,x_2,x_3) = -C_F(C_A-2C_F)\left\{\frac{2x_3}{\bar{x}_1^2}-\frac{1+x_3}{\bar{x}_1}+\left(2\log\bar{x}_1-\log x_3\right)\frac{1+x_3^2}{2\bar{x}_1\bar{x}_2}\right\}. \tag{129}$$

The evolution kernels for the remaining channels again obey active-spectator symmetry and are given by

$$P^{(1)}_{qq',q}(x_1,x_2,x_3) = P^{(1)}_{q'q,q}(x_2,x_1,x_3) = P^{(1)}_{q'\bar{q}',q}(x_2,x_3,x_1) \tag{130}$$

and

$$P^{v\,(1)}_{q\bar{q},q}(x_1,x_2,x_3) = P^{v\,(1)}_{qq,q}(x_1,x_3,x_2). \tag{131}$$

**Terms originating from LO kernels**  As specified in (50), the kernels $W^{[2,1]}$ and $W^{[2,2]}$ can be constructed from $P^{(1)}_s$ and from additional terms that originate from LO kernels. For LO channels, we find that these additional terms have the form

$$W_{a_1a_2,a_0}: \frac{\beta_0}{2}K_{a_1a_2,a_0}+P^{(0)}_{a_1a_1}\underset{1}{\otimes}K_{a_1a_2,a_0}+P^{(0)}_{a_2a_2}\underset{2}{\otimes}K_{a_1a_2,a_0}-K_{a_1a_2,a_0}\underset{12}{\otimes}P^{(0)}_{a_0a_0}, \tag{132}$$

where $K=W^{[1,0]}$ or $P^{(0)}_s$. Note that repeated parton indices on the r.h.s. are *not* summed over. The flavour diagonal DGLAP kernels $P^{(0)}_{a_ia_i}$ in (132) contain distribution terms, which we wish to make explicit. To this end, we write the kernels in the form

$$P^{(0)}_{gg}(x)=P^{(0)}_{gg,\text{reg}}(x)+\frac{2C_A}{[1-x]_+}+\frac{\beta_0}{2}\delta(1-x),$$
$$P^{(0)}_{qq}(x)=P^{(0)}_{qq,\text{reg}}(x)+\frac{2C_F}{[1-x]_+}+\frac{3}{2}C_F\delta(1-x) \tag{133}$$

in analogy to (110), where the regular terms read

$$P^{(0)}_{gg,\text{reg}}(x)=2C_A\left[\frac{1-x}{x}+x(1-x)-1\right], \qquad P^{(0)}_{qq,\text{reg}}(x)=-C_F(1+x). \tag{134}$$

We then have convolutions

$$P^{(0)}_{gg}\underset{1}{\otimes}K=P^{(0)}_{gg,\text{reg}}\left(\frac{x_1}{\bar{x}_2}\right)\frac{K(\bar{x}_2)}{\bar{x}_2}+\frac{2C_A}{[x_3]_+}K(\bar{x}_2)+\frac{1}{2}\left(\beta_0-4C_A\log x_1\right)K(x_1)\delta(x_3),$$
$$P^{(0)}_{gg}\underset{2}{\otimes}K=P^{(0)}_{gg,\text{reg}}\left(\frac{x_2}{\bar{x}_1}\right)\frac{K(x_1)}{\bar{x}_1}+\frac{2C_A}{[x_3]_+}K(x_1)+\frac{1}{2}\left(\beta_0-4C_A\log\bar{x}_1\right)K(x_1)\delta(x_3),$$
$$K\underset{12}{\otimes}P^{(0)}_{gg}=K\left(\frac{x_1}{\bar{x}_3}\right)\frac{P^{(0)}_{gg,\text{reg}}(\bar{x}_3)}{\bar{x}_3}+\frac{2C_A}{[x_3]_+}\frac{1}{\bar{x}_3}K\left(\frac{x_1}{\bar{x}_3}\right)+\frac{\beta_0}{2}K(x_1)\delta(x_3), \tag{135}$$

and

$$P^{(0)}_{qq}\underset{1}{\otimes}K=P^{(0)}_{qq,\text{reg}}\left(\frac{x_1}{\bar{x}_2}\right)\frac{K(\bar{x}_2)}{\bar{x}_2}+\frac{2C_F}{[x_3]_+}K(\bar{x}_2)+\frac{1}{2}C_F\left(3-4\log x_1\right)K(x_1)\delta(x_3),$$
$$P^{(0)}_{qq}\underset{2}{\otimes}K=P^{(0)}_{qq,\text{reg}}\left(\frac{x_2}{\bar{x}_1}\right)\frac{K(x_1)}{\bar{x}_1}+\frac{2C_F}{[x_3]_+}K(x_1)+\frac{1}{2}C_F\left(3-4\log\bar{x}_1\right)K(x_1)\delta(x_3),$$
$$K\underset{12}{\otimes}P^{(0)}_{qq}=K\left(\frac{x_1}{\bar{x}_3}\right)\frac{P^{(0)}_{qq,\text{reg}}(\bar{x}_3)}{\bar{x}_3}+\frac{2C_F}{[x_3]_+}\frac{1}{\bar{x}_3}K\left(\frac{x_1}{\bar{x}_3}\right)+\frac{3}{2}C_F K(x_1)\delta(x_3), \tag{136}$$

where we have omitted the parton labels on $K$ for brevity. To obtain these relations, we used equation (B13) in [88] to bring all plus distribution terms into the form $1/[x_3]_+$.

The convolution terms that appear in the different NLO channels are

$$
\begin{aligned}
W_{qg,g} &: P_{qg}^{(0)} \underset{1}{\otimes} K_{gg,g} + P_{gq}^{(0)} \underset{2}{\otimes} K_{q\bar{q},g} - K_{qg,q} \underset{12}{\otimes} P_{qg}^{(0)}, \\
W_{gg,q} &: P_{gq}^{(0)} \underset{1}{\otimes} K_{qg,q} + P_{gq}^{(0)} \underset{2}{\otimes} K_{gq,q} - K_{gg,g} \underset{12}{\otimes} P_{gq}^{(0)},
\end{aligned}
\tag{137}
$$

and

$$
\begin{aligned}
W_{q_j \bar{q}_k, q_i} &: \qquad\qquad \delta_{ij} P_{qg}^{(0)} \underset{2}{\otimes} K_{qg,q} - \delta_{jk} K_{q\bar{q},g} \underset{12}{\otimes} P_{gq}^{(0)}, \\
W_{q_j q_k, q_j} &: \delta_{jk} P_{qg}^{(0)} \underset{1}{\otimes} K_{gq,q} + P_{qg}^{(0)} \underset{2}{\otimes} K_{qg,q}.
\end{aligned}
\tag{138}
$$

In all these cases, only the flavour non-diagonal DGLAP kernels

$$
P_{qg}^{(0)}(x) = P_{q\bar{q},g}^{(0)}(x), \qquad\qquad P_{gq}^{(0)}(x) = P_{gq,q}^{(0)}(x)
\tag{139}
$$

appear, after we used charge conjugation symmetry to replace $P_{\bar{q}g}^{(0)}$ and $P_{g\bar{q}}^{(0)}$.

We note that for all LO and NLO channels $a_0 \to a_1 a_2$, there is exactly one parton combination in each type of convolution term. This will no longer hold for kernels at order $\alpha_s^3$, where for given $a_0$, $a_1$, and $a_2$ there is more than one possibility for the spectator partons.

In contrast to what we observed for the kernels $P_s^{(1)}$, active-spectator symmetry does *not* hold for the LO induced terms just discussed, and therefore this symmetry does not hold for $W^{[2,1]}$ and $W^{[2,2]}$ either. This is most easily seen for $W_{gg,g}^{[2,2]}$. Apart from distribution terms, all functions in the convolutions (38) are equal to $2C_A p_{gg}$ in that case. The exchange $x_1 \leftrightarrow x_3$ then leaves the first term in (38) invariant, whilst interchanging the second and third terms. Because the latter enter $W_{gg,g}^{[2,2]}$ with opposite signs in (50), active-spectator symmetry is broken.

**Nonlogarithmic terms.** The full expressions of the kernels $W^{[2,0]}$, which are not accompanied by logarithms in $W^{(2)}$, are rather lengthy, and we only discuss their general features here. Their full expressions are given in the ancillary files associated with this paper on https://arxiv.org. We note that the functions $W^{[2,0]}$ are not needed for the position space matching kernels $V^{(2)}$, but they do appear in the matching kernels $U^{(2)}$ between momentum and position space DPDs.

The general form of the $W^{[2,0]}$ coefficients can be written as

$$
\begin{aligned}
W_{gg,g}^{[2,0]} &: C_A^2 \left[ R_{gg,g}^A + \delta(x_3) D_{gg,g}^A \right] + \beta_0 C_A \delta(x_3) D_{gg,g}^\beta, \\
W_{q\bar{q},g}^{[2,0]} &: C_A T_F \left\{ R_{q\bar{q},g}^A + \delta(x_3) D_{q\bar{q},g}^A + \frac{S_{q\bar{q},g}^A}{[x_3]_+} \right\} + C_F T_F \left\{ R_{q\bar{q},g}^F + \delta(x_3) D_{q\bar{q},g}^F + \frac{S_{q\bar{q},g}^F}{[x_3]_+} \right\}, \\
W_{qg,q}^{[2,0]} &: C_A C_F \left[ R_{qg,q}^A + \delta(x_3) D_{qg,q}^A \right] + C_F^2 \left[ R_{qg,q}^F + \delta(x_3) D_{qg,q}^F \right] + \beta_0 C_F \delta(x_3) D_{qg,q}^\beta, \\
W_{qg,g}^{[2,0]} &: C_A T_F R_{qg,g}^A + C_F T_F R_{qg,g}^F, \\
W_{gg,q}^{[2,0]} &: C_A C_F R_{gg,q}^A + C_F^2 R_{gg,q}^F,
\end{aligned}
\tag{140}
$$

and

$$
\begin{aligned}
W_{qq',q}^{[2,0]} &: C_F T_F R_{qq',q}^F, & W_{q\bar{q}',q}^{[2,0]} &: C_F T_F R_{q'\bar{q}',q}^F, \\
W_{qq,q}^{\nu[2,0]} &: C_F (C_A - 2C_F) R_{qq,q}^F, & W_{q\bar{q},q}^{\nu[2,0]} &: C_F (C_A - 2C_F) R_{q\bar{q},q}^F,
\end{aligned}
\tag{141}
$$

where the functions $R^c_{a_1 a_2, a_0}$ with $c = A, F, \beta$ are regular as specified below (110). They are independent of colour factors and of the number $n_F$ of active quark flavours. We see that the colour structure of $W^{[2,0]}_{a_1 a_2, a_0}$ is the same as for the corresponding kernel $P^{(1)}_{a_1 a_2, a_0}$.

The regular parts $R^c$ are by far the most lengthy ones, containing rational functions as well as the product of rational functions with logarithms, with products of two logarithms, and with dilogarithms. The logarithms have arguments $x_1, \bar{x}_1, x_2, \bar{x}_2, x_3,$ or $\bar{x}_3$. The arguments of the dilogarithms can be reduced to the set $x_1, x_2, \bar{x}_3, x_1/\bar{x}_2, x_2/\bar{x}_1,$ and $-x_2/x_1$, using the relations given in section II.B of [96]. Note that with these arguments, all logarithms and dilogarithms are real valued. In the denominators of rational functions, the highest powers of $x_i$ and $\bar{x}_i$ (with $i = 1, 2, 3$) are the same as in the evolution kernels $P^{(1)}_s$ given earlier.

The plus distribution parts $S^c$ of the $W^{[2,0]}$ coefficients are either zero or simple polynomials in $x_1$ and $\bar{x}_1$. They determine the leading behaviour in the limit $x_1 + x_2 \to 1$ and will be explicitly given in (151) below.

The delta distribution parts $D^c$ are in general less lengthy than the regular parts, but they involve rational functions and the product of rational functions with trilogarithms, with products of logarithms and dilogarithms, with dilogarithms, and with products of up to three logarithms. The arguments of all logarithms and polylogarithms are only $x_1$ or $\bar{x}_1$. The rational functions often coincide with the corresponding functions in (115), or at least they have the same denominators as these functions.

## 5.1 Parton number and momentum sum rules

Sum rules for momentum space DPDs at $\Delta = 0$ were proposed in [53]. They are among the few theoretical constraints on DPDs that we have. A detailed proof of these sum rules was given in [54], where it was also shown that the renormalisation scale independence of the DPD sum rules implies corresponding sum rules for the $1 \to 2$ evolution kernels. These read

$$\int\limits_0^{1-x_1} dx_2 \left[ P_{a_1 q, a_0}(x_1, x_2) - P_{a_1 \bar{q}, a_0}(x_1, x_2) \right] = \left( \delta_{a_1 \bar{q}} - \delta_{a_1 q} - \delta_{a_0 \bar{q}} + \delta_{a_0 q} \right) P_{a_1 a_0}(x_1) \quad (142)$$

and

$$\sum_{a_2} \int\limits_0^{1-x_1} dx_2 \, x_2 \, P_{a_1 a_2, a_0}(x_1, x_2) = (1 - x_1) P_{a_1 a_0}(x_1). \quad (143)$$

Note that the distribution terms in the DGLAP evolution kernels on the r.h.s. cancel. In the number sum rule (142), such terms only appear if $a_0 = a_1$, in which case the sum of Kronecker deltas gives zero. In the momentum sum rule (143), distribution terms are removed by the prefactor $1 - x_1$.

For the LO kernels, the sum rules (142) and (143) are readily verified using the relation (37) and the list of possible transitions $a_0 \to a_1 a_2$. At NLO, however, the sum rules provide nontrivial relations between the evolution kernels $P^{(1)}_s$ and the DGLAP splitting functions, which are well known at that order [44–52]. These relations serve as a valuable cross check of our results.

**Number sum rule.** Let us first consider the number sum rule (142) for the different parton combinations. Starting with the ones that involve $1 \to 2$ evolution kernels in LO channels, we

have

$$\int dx_2 \, P^{(1)}_{q\bar{q},g} = P^{(1)}_{qg},$$

$$\int dx_2 \, P^{(1)}_{gq,q} = P^{(1)}_{gq}, \tag{144}$$

where the arguments of the kernels and integration boundaries are as in (142). Here and in the following we omit NLO kernels that vanish identically, such as $P^{(1)}_{qq,g}$ and $P^{(1)}_{g\bar{q},q}$. Turning to the NLO channels, we use the notation

$$P_{q_i q_k} = \delta_{ik} P^V_{qq} + P^S_{qq},$$
$$P_{\bar{q}_i q_k} = \delta_{ik} P^V_{\bar{q}q} + P^S_{\bar{q}q} \tag{145}$$

for DGLAP kernels [97] and note that at NLO the relation

$$P^S_{qq} = P^S_{\bar{q}q} \tag{146}$$

holds for the flavour singlet parts. Taking linear combinations of the number sum rules for the different transitions $q_j \to q_j q_k$ and $q_i \to q_j \bar{q}_k$ and using the symmetry relations (114), we obtain

$$\int dx_2 \, P^{(1)v}_{qq,q} = \int dx_2 \, P^{(1)v}_{q\bar{q},q},$$

$$\int dx_2 \, P^{(1)v}_{\bar{q}q,q} = 2 P^{V\,(1)}_{\bar{q}q},$$

$$\int dx_2 \, P^{(1)}_{q'q,q} = \int dx_2 \, P^{(1)}_{q'\bar{q}',q} = P^{S\,(1)}_{qq}. \tag{147}$$

We checked explicitly that our results for the $1 \to 2$ kernels fulfil all of the above sum rules.

**Momentum sum rule.** For NLO kernels, one finds the momentum sum rules

$$\int dx_2 \, x_2 \left[ P^{(1)}_{gg,g} + 2n_f P^{(1)}_{gq,g} \right] = (1-x_1) P^{(1)}_{gg},$$

$$\int dx_2 \, x_2 \left[ P^{(1)}_{gq,q} + P^{(1)}_{gg,q} \right] = (1-x_1) P^{(1)}_{gq},$$

$$\int dx_2 \, x_2 \left[ P^{(1)}_{q\bar{q},g} + P^{(1)}_{qg,g} \right] = (1-x_1) P^{(1)}_{qg},$$

$$\int dx_2 \, x_2 \left[ P^{(1)}_{qg,q} + P^{(1)v}_{qq,q} + P^{(1)v}_{q\bar{q},q} + 2n_f P^{(1)}_{qq',q} \right] = (1-x_1) P^{V\,(1)}_{qq},$$

$$\int dx_2 \, x_2 \, P^{(1)v}_{\bar{q}q,q} = (1-x_1) P^{V\,(1)}_{\bar{q}q},$$

$$\int dx_2 \, x_2 \left[ P^{(1)}_{q'q,q} + P^{(1)}_{q'\bar{q}',q} \right] = (1-x_1) P^{S\,(1)}_{qq}, \tag{148}$$

all of which are fulfilled for the $1 \to 2$ evolution kernels we have calculated.

## 5.2 Active-spectator symmetry

In this subsection, we consider kernels away from the kinematic point $x_3 = 0$, so that only real emission graphs contribute and distribution terms do not contribute. We find active-spectator symmetry to hold for the evolution kernels $P_s^{(1)}$ in all parton channels, but not for the any of the kernels $W^{[2,k]}$ that make up the matching coefficient $W^{(2)}$.

The lack of this symmetry for $W^{(2)}$ is not surprising, because the renormalisation (26) of $W$ clearly treats observed and spectator partons in an asymmetric way. Furthermore, already in bare graphs the active partons are singled out because their transverse momenta differ by $\pm \Delta$ in the amplitude and its conjugate. This is in fact the only difference between active partons and spectators. To see this, we consider the bare DPDs $F_B$ in light-cone perturbation theory, which can be obtained from covariant perturbation theory by integrating over all minus momenta (see e.g. chapter 7.2.3 of [70] and references therein). Both active and spectator partons are then on their mass shell by construction. As shown in section 5 of [54], for $\Delta = 0$ the numerator factors in light-cone perturbation theory are also identical for active and spectator partons in bare DPDs, provided of course that one considers unpolarised partons.

To understand why active-spectator symmetry holds for $P_{a_1 a_2, a_0}^{(1)}(x_1, x_2, x_3)$, we note that for $x_3 > 0$ these evolution kernels are associated with the ultraviolet divergences of two-loop graphs for the splitting process $a_0 \to a_1 a_2 a_3$, where $a_3$ is the spectator parton. More precisely, these kernels arise from kinematic configurations in which all three final state partons have large transverse momenta. The finite value of $\Delta$ can be neglected in that region, so that one should have active-spectator symmetry according to our arguments in the previous paragraph. Note that divergences from configurations in which only two of the three final state partons have transverse momenta in the ultraviolet are not associated with $P_s^{(1)}$, but with the lower-order splitting kernels $P_s^{(0)}$ or $P^{(0)}$.

Let us give another reason why the finite transverse momentum $\Delta$ in our calculation should play no role for the kernels $P_s$. We obtained $P_s^{(1)}$ from the computation of the splitting kernels $W(\Delta)$, where $\Delta$ plays the role of a hard scale and cannot be set to zero. On the other hand, the evolution kernels $P_s$ appear in the evolution equation of the DPDs $F(\Delta)$, which is valid at $\Delta = 0$. One could hence compute $P_s^{(1)}$ from graphs with $\Delta = 0$, provided that one has a dimensionful infrared regulator for separating infrared and ultraviolet poles in dimensional regularisation.

## 5.3 Kinematic limits

In this subsection, we analyse the behaviour of the kernels $P_s^{(1)}$ and $W^{(2)}$ in various kinematic limits.

### 5.3.1 Threshold limit: large $x_1 + x_2$

To begin with, we consider the convolution (112) at the parton-level threshold $x_1 + x_2 \to 1$, i.e. $x_3 \to 0$. Writing out the plus-distribution, we readily obtain

$$K \underset{12}{\otimes} f \underset{x_3 \to 0}{=} -\left[ K_p(x_1, x_2) \log x_3 - K_\delta(x_1) \right] \frac{f(x_1 + x_2)}{x_1 + x_2}. \tag{149}$$

A logarithm in $x_3$ is thus generated by the plus distribution term of the kernel $K$, which hence dominates the threshold behaviour of the convolution. Since the kernels do not contain distributions $\left[ x_3^{-1} \log^k x_3 \right]_+$ with $k > 0$, no higher powers of $\log x_3$ are generated in the convolution (112). This situation is analogous to the convolution of ordinary DGLAP splitting functions with PDFs, where no such distributions appear even at NNLO [98, 99].

The plus distribution parts of the LO channel kernels are given by

$$P^{(1)}_{q\bar{q},g;p} = -2(C_A - 2C_F)T_F (x_1 \bar{x}_1 + x_2 \bar{x}_2),$$

$$P^{(1)}_{qg,q;p} = 2C_A C_F \bar{x}_1, \tag{150}$$

and

$$W^{[2,2]}_{gg,g;p} = C_A^2 \left[ p_{gg}(x_1) + p_{gg}(x_2) \right],$$

$$W^{[2,2]}_{q\bar{q},g;p} = -\frac{(C_A - 2C_F)T_F}{2} \left[ p_{qg}(x_1) + p_{qg}(x_2) \right], \qquad W^{[2,0]}_{q\bar{q},g;p} = \frac{1}{2} P^{(1)}_{q\bar{q},g;p},$$

$$W^{[2,2]}_{qg,q;p} = C_A C_F \, p_{qq}(x_1), \tag{151}$$

where we omitted coefficients $P^{(1)}_{a_1 a_2, a_0; p}$ and $W^{[2,k]}_{a_1 a_2, a_0; p}$ that are exactly zero.

The kernels $P^{(1)}_s$ and $W^{(2)}$ for NLO channels have no plus distribution terms. Their leading threshold behaviour is $\log^n x_3$ with $n = 0, 1, 2$, which gives a convolution $K \otimes f$ of order $x_3 \log^n x_3$. Specifically, we have

$$P^{(1)}_{gg,g} \sim \mathcal{O}(\log x_3), \qquad P^{(1)}_{qg,g} \sim \mathcal{O}(\log x_3), \qquad P^{(1)}_{gg,q} \sim \mathcal{O}(\log x_3),$$

$$P^{(1)}_{qq',q} \sim \mathcal{O}(\log x_3), \qquad P^{(1)}_{q'\bar{q}',q} \sim \mathcal{O}(\log x_3), \qquad P^{v(1)}_{qq,q} \sim \mathcal{O}(\log x_3),$$

$$P^{v(1)}_{q\bar{q},q} \sim \mathcal{O}(1), \tag{152}$$

and

$$\begin{aligned}
W^{[2,0]}_{qg,g} &\sim \mathcal{O}(\log^2 x_3), & W^{[2,1]}_{qg,g} &\sim \mathcal{O}(\log x_3), & W^{[2,2]}_{qg,g} &\sim \mathcal{O}(1), \\
W^{[2,0]}_{gg,q} &\sim \mathcal{O}(\log^2 x_3), & W^{[2,1]}_{gg,q} &\sim \mathcal{O}(\log x_3), & W^{[2,2]}_{gg,q} &\sim \mathcal{O}(1), \\
W^{[2,0]}_{qq',q} &\sim \mathcal{O}(\log^2 x_3), & W^{[2,1]}_{qq',q} &\sim \mathcal{O}(\log x_3), & W^{[2,2]}_{qq',q} &\sim \mathcal{O}(1), \\
W^{[2,0]}_{q'\bar{q}',q} &\sim \mathcal{O}(\log^2 x_3), & W^{[2,1]}_{q'\bar{q}',q} &\sim \mathcal{O}(\log x_3), & W^{[2,2]}_{q'\bar{q}',q} &= 0, \\
W^{v[2,0]}_{qq,q} &\sim \mathcal{O}(\log^2 x_3), & W^{v[2,1]}_{qq,q} &\sim \mathcal{O}(\log x_3), & W^{v[2,2]}_{qq,q} &= 0, \\
W^{v[2,0]}_{q\bar{q},q} &\sim \mathcal{O}(\log x_3), & W^{v[2,1]}_{q\bar{q},q} &\sim \mathcal{O}(1), & W^{v[2,2]}_{q\bar{q},q} &= 0.
\end{aligned} \tag{153}$$

### 5.3.2 Small $x_1 + x_2$

We now turn to the case in which both momentum fractions $x_1$ and $x_2$ are small; this is a typical situation at high collision energy when two systems of moderately large invariant mass are produced. In the limit $x_1 + x_2 \to 0$ the convolution integral (112) includes the region $z \to 0$ in the kernel $K(zu, z\bar{u})$, where $u$ is defined in terms of the external momentum fractions $x_1$ and $x_2$ by (113).

Let us first recall the analogous situation for the one-dimensional Mellin convolution of an LO DGLAP kernel and a PDF,

$$P^{(0)} \otimes f = \int_x^1 \frac{dz}{z} P^{(0)}(z) f\left(\frac{x}{z}\right). \tag{154}$$

If $P(x) \propto x^{-1}$ and $f(x) \propto x^{-1} \log^k x^{-1}$ for $x \ll 1$, then the region $x \ll z \ll 1$ in (154) gives a behaviour proportional to

$$\frac{1}{x} \int \frac{dz}{z} \log^k \frac{z}{x} \sim \frac{1}{k+1} \frac{1}{x} \log^{k+1} \frac{1}{x}, \tag{155}$$

where we obtained the leading logarithmic behaviour in $x$ on the r.h.s. by extending the integration to the full range $x \le z \le 1$. This is nothing but the generation of small-$x$ logarithms in a PDF: starting with $k$ powers of $\log x^{-1}$, the convolution with a kernel proportional to $1/x$ results in an additional power of that logarithm.

In full analogy one obtains a small-$x$ logarithm in the two-variable convolution (112) if $f(x)$ has the same small-$x$ behaviour as above and

$$K(zu, z\bar{u}) \sim \frac{w(u)}{z^2} \qquad \text{for } z \ll 1. \tag{156}$$

The region $x_1 + x_2 \ll z \ll 1$ then contributes to $K \otimes f$ as

$$\frac{w(u)}{(x_1 + x_2)^2} \int \frac{dz}{z} \log^k \frac{z}{x_1 + x_2} \sim \frac{w(u)}{k+1} \frac{1}{(x_1 + x_2)^2} \log^{k+1} \frac{1}{x_1 + x_2}, \tag{157}$$

where on the r.h.s. we extracted the leading logarithmic behaviour by extending the integration to the full range $x_1 + x_2 \le z \le 1$. If $K(zu, z\bar{u})$ grows less fast than $z^{-2}$ at small $z$, then the region $z \ll 1$ is no longer dominant in the convolution integral, and no small-$x$ logarithm is built up by the integration over $z$. Notice also that the $u$ dependence of the kernel in (156) directly determines the $u$ dependence of the convolution (157).

For the $1 \to 2$ evolution kernels $P_s^{(1)}(zu, z\bar{u})$, we find the following leading behaviour at small $z$:

$$P_{gg,g}^{(1)} \sim \frac{2C_A^2}{z^2} \left[ 1 - 6u\bar{u} + \frac{2u^3 - 2u^2 + 4u - 1}{u} \log \bar{u} + \frac{2\bar{u}^3 - 2\bar{u}^2 + 4\bar{u} - 1}{\bar{u}} \log u \right],$$

$$P_{q\bar{q},g}^{(1)} \sim -\frac{2C_A T_F}{z^2} \left[ p_{qg}(u) \log(u\bar{u}) + (1 - 2u)^2 \right],$$

$$P_{gg,q}^{(1)} \sim \frac{C_F}{C_A} P_{gg,g}^{(1)},$$

$$P_{q'\bar{q}',q}^{(1)} \sim \frac{C_F}{C_A} P_{q\bar{q},g}^{(1)}. \tag{158}$$

For the kernels without a $z^{-2}$ singularity, we find

$$\begin{aligned}
P_{qg,q}^{(1)} &\sim \mathcal{O}(z^{-1}), & P_{qg,g}^{(1)} &\sim \mathcal{O}(z^{-1}), \\
P_{qq',q}^{(1)} &\sim \mathcal{O}(1), & P_{qq,q}^{v\,(1)} &\sim \mathcal{O}(z^2), & P_{q\bar{q},q}^{v\,(1)} &\sim \mathcal{O}(z^{-1}).
\end{aligned} \tag{159}$$

For the $W^{[2,k]}$ coefficients going like $z^{-2}$ at small $z$, we find

$$W_{gg,g}^{[2,0]} \sim \frac{C_A^2}{z^2}\Bigg[3 - 20u\bar{u} - \frac{2u^3 - 2u^2 + 4u - 3}{6u}\pi^2 + (1 - 6u\bar{u})\log(u\bar{u})$$

$$+ \frac{2u^3 - 2u^2 + 4u - 1}{2u}\log^2\bar{u} - \frac{2u^4 - 4u^3 + 6u^2 + 3u - 3}{2u\bar{u}}\log^2 u$$

$$- \frac{2u^4 - 4u^3 + 6u^2 - 7u + 2}{u\bar{u}}\log u\log\bar{u} + \frac{3(1 - 2u)}{u\bar{u}}\mathrm{Li}_2\frac{u-1}{u}\Bigg],$$

$$W_{gg,g}^{[2,1]} \sim \frac{2C_A^2}{z^2}\Bigg[1 - 6u\bar{u} + \frac{2u^3 - 2u^2 + 4u - 1}{u}\log\bar{u} + \frac{2\bar{u}^3 - 2\bar{u}^2 + 4\bar{u} - 1}{\bar{u}}\log u\Bigg],$$

$$W_{gg,g}^{[2,2]} \sim -\frac{2C_A^2}{z^2}\frac{u^4 - 2u^3 + 3u^2 - 2u - 1}{u\bar{u}},$$

$$W_{q\bar{q},g}^{[2,0]} \sim \frac{C_A T_F}{6z^2}\Big[p_{qg}(u)\big(\pi^2 - 72 - 3\log^2(u\bar{u}) - 24\log(u\bar{u})\big) + 48 + 18\log(u\bar{u})\Big],$$

$$W_{q\bar{q},g}^{[2,1]} \sim \frac{2C_A T_F}{z^2}\Big[2 - p_{qg}(u)\big(\log(u\bar{u}) + 3\big)\Big],$$

$$W_{q\bar{q},g}^{[2,2]} \sim -\frac{C_A T_F}{z^2}p_{qg}(u), \tag{160}$$

and

$$W_{gg,q}^{[2,0]} \sim \frac{C_F}{C_A}W_{gg,g}^{[2,0]}, \qquad W_{gg,q}^{[2,1]} \sim \frac{C_F}{C_A}W_{gg,g}^{[2,1]}, \qquad W_{gg,q}^{[2,2]} \sim \frac{2}{z^2}\Bigg[\frac{2C_F^2}{u\bar{u}} - C_A C_F\, p_{gg}(u)\Bigg],$$

$$W_{q'\bar{q}',q}^{[2,0]} \sim \frac{C_F}{C_A}W_{q\bar{q},g}^{[2,0]}, \qquad W_{q'\bar{q}',q}^{[2,1]} \sim \frac{C_F}{C_A}W_{q\bar{q},g}^{[2,1]}, \qquad W_{q'\bar{q}',q}^{[2,2]} = 0, \tag{161}$$

while for the coefficients with a less singular behaviour we have

$$\begin{array}{lll}
W_{qg,q}^{[2,0]} \sim \mathcal{O}(z^{-1}), & W_{qg,q}^{[2,1]} \sim \mathcal{O}(z^{-1}), & W_{qg,q}^{[2,2]} \sim \mathcal{O}(z^{-1}), \\[4pt]
W_{qg,g}^{[2,0]} \sim \mathcal{O}(z^{-1}), & W_{qg,g}^{[2,1]} \sim \mathcal{O}(z^{-1}), & W_{qg,g}^{[2,2]} \sim \mathcal{O}(z^{-1}), \\[4pt]
W_{q'q,q}^{[2,0]} \sim \mathcal{O}(1), & W_{q'q,q}^{[2,1]} \sim \mathcal{O}(1), & W_{q'q,q}^{[2,2]} \sim \mathcal{O}(1), \\[4pt]
W_{qq,q}^{\nu[2,0]} \sim \mathcal{O}(1), & W_{qq,q}^{\nu[2,1]} \sim \mathcal{O}(z^2), & W_{qq,q}^{\nu[2,2]} = 0, \\[4pt]
W_{q\bar{q},q}^{\nu[2,0]} \sim \mathcal{O}(z^{-1}), & W_{q\bar{q},q}^{\nu[2,1]} \sim \mathcal{O}(z^{-1}), & W_{q\bar{q},q}^{\nu[2,2]} = 0.
\end{array} \tag{162}$$

For both types of kernels, the channels with a $z^{-2}$ behaviour are $g \to gg$, $g \to q\bar{q}$, $q \to gg$, and $q \to q'\bar{q}'$. We checked that the graphs giving rise to such a behaviour are those in which — on both sides of the final-state cut — the spectator parton is emitted from a three-particle vertex at which a slow gluon is emitted, where "slow" is relative to the incoming parton at that vertex. Examples for such graphs are figures 1a to 1d and figures 2g to 2j.

Notice finally the Casimir scaling between $q \to gg$ and $g \to gg$, and between $q \to q'\bar{q}'$ and $g \to q\bar{q}$. The small $z$ limits of $P_s^{(1)}$, $W^{[2,0]}$, and $W^{[2,1]}$ (but not of $W^{[2,2]}$) in these pairs of channels are simply related by a proportionality factor $C_F/C_A$.

**Triple Regge limit.** The results just presented also allow us to analyse the "triple Regge limit" $x_1 \ll x_1 + x_2 \ll 1$. To this end, we simply take the limit $u \to 0$ in (157). Restricting our attention to the kernels with a $z^{-2}$ behaviour, we have

$$P_{gg,g}^{(1)} \sim \frac{2C_A^2}{z^2}(2 + 3\log u), \qquad\qquad P_{gg,q}^{(1)} \sim \frac{C_F}{C_A}P_{gg,g}^{(1)},$$

$$P_{q\bar{q},g}^{(1)} \sim -\frac{2C_A T_F}{z^2}(1 + \log u), \qquad\qquad P_{q'\bar{q}',q}^{(1)} \sim \frac{C_F}{C_A}P_{q\bar{q},g}^{(1)}, \tag{163}$$

and

$$W_{gg,g}^{[2,0]} \sim \frac{C_A^2}{6z^2}\left(36 - \pi^2 + 9\log^2 u\right),$$

$$W_{gg,g}^{[2,1]} \sim \frac{2C_A^2}{z^2}\left(2 + 3\log u\right), \qquad\qquad W_{gg,g}^{[2,2]} \sim \frac{2C_A^2}{z^2}\frac{1}{u},$$

$$W_{q\bar{q},g}^{[2,0]} \sim \frac{C_A T_F}{6z^2}\left(\pi^2 - 24 - 6\log u - 3\log^2 u\right),$$

$$W_{q\bar{q},g}^{[2,1]} \sim -\frac{2C_A T_F}{z^2}\left(1 + \log u\right), \qquad\qquad W_{q\bar{q},g}^{[2,2]} \sim -\frac{C_A T_F}{z^2}, \qquad (164)$$

and

$$W_{gg,q}^{[2,0]} \sim \frac{C_F}{C_A} W_{gg,g}^{[2,0]}, \qquad W_{gg,q}^{[2,1]} \sim \frac{C_F}{C_A} W_{gg,g}^{[2,1]}, \qquad W_{gg,q}^{[2,2]} \sim -\frac{2C_F(C_A - 2C_F)}{z^2}\frac{1}{u},$$

$$W_{q'\bar{q}',q}^{[2,0]} \sim \frac{C_F}{C_A} W_{q\bar{q},g}^{[2,0]}, \qquad W_{q'\bar{q}',q}^{[2,1]} \sim \frac{C_F}{C_A} W_{q\bar{q},g}^{[2,1]}, \qquad W_{q'\bar{q}',q}^{[2,2]} = 0. \qquad (165)$$

We see that the leading singular behaviour of all these kernels involves at most two powers of $\log u \approx \log(x_1/x_2)$. An exception are the kernels $W_{gg,a_0}^{[2,2]}$ for the emission of two gluons. In this case we have a power-law behaviour like $u^{-1} \approx x_2/x_1$ of the kernels, which results in a power-law behaviour like $u^{-1}(x_1 + x_2)^{-2} \approx (x_1 x_2)^{-1}$ of the convolution (157). One finds that this behaviour comes from the terms $P_{ga_0}^{(0)} \underset{1}{\otimes} P_{a_0 g,g}^{(0)}$ and $P_{ga_0}^{(0)} \underset{2}{\otimes} P_{ga_0,g}^{(0)}$ and corresponds to graphs with UD topology as shown in figures 1b and 2h. In these graphs there are two consecutive three-point vertices at which an observed slow gluon is radiated from a parton carrying the full or almost the full initial plus-momentum. The appearance of a $1/u$ power behaviour only in $W^{[2,2]}$ but not in $W^{[2,1]}$ or $P_s^{(1)}$ means that this behaviour goes along with a double logarithm $\log^2(\mu^2/\Delta^2)$. This corresponds to strong ordering in the transverse momenta at the two consecutive splitting vertices, which gives two logarithmic transverse-momentum integrals.

Analogous expressions for limit $x_2 \ll x_1 + x_2$ are easy to obtain, because the channels with a $z^{-2}$ behaviour have kernels $K(u_1, u_2)$ that are symmetric in $u_1$ and $u_2$.

### 5.3.3 Small $x_1$ or $x_2$

We now consider the limit in which $x_1 \ll 1$ whilst $x_2$ is not. This is relevant for DPS processes in which one of the two hard systems is very heavy, whereas the other one is light compared with the total collision energy. The integration in the convolution (112) can in this case not reach the region $z \ll 1$. Hence, the $z$ integration cannot build up a small-$x$ logarithm, and the limit $x_1 \ll 1$ of the convolution is simply obtained from the limit $u \ll 1$ in the kernel $K(zu, z\bar{u})$. Note that, unlike in the case $x_1 + x_2 \ll 1$, the distribution parts $K_p$ and $K_\delta$ of the kernel can also contribute to the small $x_1$ behaviour.

In the following, we give the behaviour of all kernels for the case in which $x_1 \ll 1$ whilst $x_2$ is unconstrained. This covers the limit just discussed, and it will also allow us to consider the nested limit $x_1 \ll x_2 \ll 1$ later on. An analogous discussion holds of course for the limit $x_2 \ll 1$ at generic values of $x_1$. For small $u$ and generic $z$, one finds for the $1 \to 2$ evolution

kernels

$$P_{gg,g}^{(1)} \sim \frac{2\delta(1-z)}{3u}\left[C_A^2(4-\pi^2)+5\beta_0\,C_A\right],$$

$$P_{gq,q}^{(1)} \sim \frac{2C_A C_F}{3u}\left[\frac{1-z}{z}+\delta(1-z)(4-\pi^2)\right]+\frac{10\beta_0\,C_F\,\delta(1-z)}{3u},$$

$$P_{gq,g}^{(1)} \sim -\frac{4(C_A-2C_F)\,T_F\,(1-z)}{u},$$

$$P_{gg,q}^{(1)} \sim \frac{2C_A C_F}{u},\tag{166}$$

while the ones without a $u^{-1}$ singularity go like

$$
\begin{array}{lll}
P_{q\bar{q},g}^{(1)} \sim \mathcal{O}(\log^2 u), & P_{qg,q}^{(1)} \sim \mathcal{O}(\log^2 u), & P_{qg,g}^{(1)} \sim \mathcal{O}(\log u), \\
P_{qq',q}^{(1)} \sim \mathcal{O}(\log u), & P_{q'q,q}^{(1)} \sim \mathcal{O}(\log u), & P_{q'\bar{q}',q}^{(1)} \sim \mathcal{O}(\log u), \\
P_{qq,q}^{v(1)} \sim \mathcal{O}(1), & P_{q\bar{q},q}^{v(1)} \sim \mathcal{O}(\log u), & P_{\bar{q}q,q}^{v(1)} \sim \mathcal{O}(\log u).
\end{array}\tag{167}
$$

The $W^{[2,k]}$ coefficients are in this limit given by

$$W_{gg,g}^{[2,0]} \sim \frac{2\delta(1-z)}{9u}\left[C_A^2\left(16-45\zeta(3)\right)+14\beta_0\,C_A\right],$$

$$W_{gg,g}^{[2,1]} \sim \frac{2\delta(1-z)}{3u}\left[C_A^2(4-\pi^2)+5\beta_0\,C_A\right],$$

$$W_{gg,g}^{[2,2]} \sim \frac{2C_A^2}{u}\left\{\frac{1}{[1-z]_+}+\frac{(1-z)(1+z^2)}{z^2}-\delta(1-z)\log u\right\}+\frac{\beta_0\,C_A\,\delta(1-z)}{u},$$

$$W_{gq,q}^{[2,0]} \sim \frac{C_F}{C_A}\,W_{gg,g}^{[2,0]},$$

$$W_{gq,q}^{[2,1]} \sim \frac{C_F}{C_A}\,W_{gg,g}^{[2,1]},$$

$$W_{gq,q}^{[2,2]} \sim \frac{2C_A C_F}{u}\left\{\frac{1}{[1-z]_+}+\frac{1-z}{2z}-\delta(1-z)\log u\right\}+\frac{\beta_0\,C_F\,\delta(1-z)}{u},\tag{168}$$

and

$$W_{gq,g}^{[2,0]} \sim -\frac{2(C_A-2C_F)\,T_F}{u}(1-z),$$

$$W_{gq,g}^{[2,1]} \sim -\frac{4(C_A-C_F)\,T_F}{u}(1-z), \qquad W_{gq,g}^{[2,2]} \sim \frac{C_A\,T_F}{u}\frac{p_{qg}(z)}{z},$$

$$W_{gg,q}^{[2,0]} \sim \mathcal{O}(\log^2 u),$$

$$W_{gg,q}^{[2,1]} \sim \frac{2C_F\,(C_A-C_F)}{u}, \qquad W_{gg,q}^{[2,2]} \sim -\frac{C_F\,(C_A-2C_F)}{u}\frac{p_{gq}(z)}{z}.\tag{169}$$

The limiting behaviour of the subleading kernels is given by

$$
\begin{aligned}
W_{q\bar{q},g}^{[2,0]} &\sim \mathcal{O}(\log^3 u), & W_{q\bar{q},g}^{[2,1]} &\sim \mathcal{O}(\log^2 u), & W_{q\bar{q},g}^{[2,2]} &\sim \mathcal{O}(\log u), \\
W_{qg,q}^{[2,0]} &\sim \mathcal{O}(\log^3 u), & W_{qg,q}^{[2,1]} &\sim \mathcal{O}(\log^2 u), & W_{qg,q}^{[2,2]} &\sim \mathcal{O}(\log u), \\
W_{qg,g}^{[2,0]} &\sim \mathcal{O}(\log^2 u), & W_{qg,g}^{[2,1]} &\sim \mathcal{O}(\log u), & W_{qg,g}^{[2,2]} &\sim \mathcal{O}(1). \\
W_{qq',q}^{[2,0]} &\sim \mathcal{O}(\log^2 u), & W_{qq',q}^{[2,1]} &\sim \mathcal{O}(\log u), & W_{qq',q}^{[2,2]} &\sim \mathcal{O}(1), \\
W_{q'q,q}^{[2,0]} &\sim \mathcal{O}(\log^2 u), & W_{q'q,q}^{[2,1]} &\sim \mathcal{O}(\log u), & W_{q'q,q}^{[2,2]} &\sim \mathcal{O}(1), \\
W_{q'\bar{q}',q}^{[2,0]} &\sim \mathcal{O}(\log^2 u), & W_{q'\bar{q}',q}^{[2,1]} &\sim \mathcal{O}(\log u), & W_{q'\bar{q}',q}^{[2,2]} &= 0, \\
W_{qq,q}^{\nu[2,0]} &\sim \mathcal{O}(\log u), & W_{qq,q}^{\nu[2,1]} &\sim \mathcal{O}(1), & W_{qq,q}^{\nu[2,2]} &= 0, \\
W_{q\bar{q},q}^{\nu[2,0]} &\sim \mathcal{O}(\log u), & W_{q\bar{q},q}^{\nu[2,1]} &\sim \mathcal{O}(1), & W_{q\bar{q},q}^{\nu[2,2]} &= 0, \\
W_{\bar{q}q,q}^{\nu[2,0]} &\sim \mathcal{O}(\log u), & W_{\bar{q}q,q}^{\nu[2,1]} &\sim \mathcal{O}(1), & W_{\bar{q}q,q}^{\nu[2,2]} &= 0.
\end{aligned}
\tag{170}
$$

We see that the channels with a $u^{-1}$ singularity are those in which the parton with momentum fraction $x_1$ is a gluon. There are hence graphs in which that slow gluon is radiated from a parton with momentum fraction much larger than $x_1$, i.e. one has a vertex with the emission of a "slow" gluon in the same sense as specified after (162).

The preceding expressions are obtained by approximating the kernels $K(zu, z\bar{u})$ for $u = x_1/(x_1 + x_2) \ll 1$. One may subsequently take the limit $z \ll 1$, which becomes relevant in the convolution if $x_1 + x_2 \ll 1$. Comparing the results of this procedure with those in (163) to (165), we find that in general the limits $u \ll 1$ and $z \ll 1$ do not commute. The only case when they do commute is if a kernel has the maximally singular behaviour $z^{-2} u^{-1}$ in both limits. This holds for the kernels $W_{gg,g}^{[2,2]}$ and $W_{gg,q}^{[2,2]}$ we already discussed after (165). In all other cases, the behaviour of the convolution $K \otimes f$ in the triple Regge limit $x_1 \ll x_1 + x_2 \ll 1$ depends on the direction in which this limit is approached.

## 5.4 Numerical illustration

A systematic investigation of the numerical impact of the NLO kernels we have computed is a substantial task, far beyond the scope of this paper. Let us, however, illustrate the impact of the NLO corrections to $1 \to 2$ splitting in a simple setting. We compute the perturbative splitting contribution to the two-gluon DPD at NLO,[6]

$$
F_{gg}^{\mathrm{NLO}}(y, \mu) = F_{gg}^{(1)}(y, \mu) + F_{gg}^{(2)}(y, \mu),
\tag{171}
$$

with

$$
F_{gg}^{(1)}(y, \mu) = \frac{a_s(\mu)}{\pi y^2} V_{gg,g}^{(1)}(y, \mu) \underset{12}{\otimes} f_g(\mu),
\tag{172}
$$

$$
F_{gg}^{(2)}(y, \mu) = \frac{a_s^2(\mu)}{\pi y^2} \left\{ V_{gg,g}^{(2)}(y, \mu) \underset{12}{\otimes} f_g(\mu) + V_{gg,q}^{(2)}(y, \mu) \underset{12}{\otimes} \sum_q \left[ f_q(\mu) + f_{\bar{q}}(\mu) \right] \right\},
\tag{173}
$$

where $a_s$ and the parton densities on the r.h.s. are evolved at NLO. We set $\mu = b_0/y$, so that only the part $V^{[2,0]} = W^{[2,1]}$ of the second-order splitting kernels contributes. We take $y = 0.022\,\mathrm{fm}$ for the transverse distance between the partons, which corresponds to $\mu = 10\,\mathrm{GeV}$. We use the CT14 PDF set [100] at NLO, with an associated value of $\alpha_s(10\,\mathrm{GeV}) = 0.178$ for the strong coupling at the scale considered here.

---

[6]Note that the label "NLO" on $F_{gg}$ refers to the sum of $\mathcal{O}(a_s)$ and $\mathcal{O}(a_s^2)$ terms.

In figure 7 we show $F^{\text{NLO}}$ along with its first-order contribution $F^{(1)}$. We also show the leading-order expression $F^{\text{LO}}$ of the two-gluon DPD, which has the same form as $F^{(1)}$ in (172) but with $a_s$ and the gluon distribution evaluated at LO. The latter are again taken from the CT14 analysis, with an LO coupling of $\alpha_s(10\,\text{GeV}) = 0.2$.[7] In figure 8 we show the ratios

$$R^{\text{NLO/LO}-1} = F^{\text{NLO}}_{gg}\big/F^{\text{LO}}_{gg} - 1\,, \qquad\qquad R^{(2)/(1)} = F^{(2)}_{gg}\big/F^{(1)}_{gg}\,. \qquad (174)$$

The first ratio quantifies the relative change when going from the LO to the NLO approximation of the DPD, whereas the second ratio indicates the relative importance of the $\mathcal{O}(a_s^2)$ kernels in the NLO results. Notice that $x_2$ in figure 8 goes up to the kinematic boundary $x_2 = 1 - x_1$.

We see that the difference between $F_{gg}$ at LO and NLO can be appreciable, especially for small momentum fractions and when $x_2$ is close to its kinematic boundary $1 - x_1$. Typically, there is a considerable difference between $F^{\text{LO}}$ and $F^{(1)}$, which simply reflects the difference between $f_g(x_1 + x_2)$ at LO and NLO. That this difference is large for the gluon density at moderately large scales $\mu$ is in fact well known and not specific to the CT14 parton distributions. Compared to this effect, the impact of the $\mathcal{O}(a_s^2)$ term $F^{(2)}$ is smaller, but nevertheless its size amounts to 5% to 10% of $F^{(1)}$ in a wide kinematic regime. We also produced analogous plots with the LO and NLO sets of MMHT 2014 [102] and of NNPDF3.0 [103]. For $x_1 + x_2$ up to 0.1, they look quite similar to those for CT14. For larger momentum fractions the differences between the sets become more pronounced, due to larger differences between the respective gluon distributions.

In figure 8 we also show the ratio

$$R^{(q)/(1)} = F^{(q)}_{gg}\big/F^{(1)}_{gg}\,, \qquad (175)$$

with

$$F^{(q)}_{gg}(y,\mu) = \frac{a_s^2(\mu)}{\pi\,y^2}\, V^{(2)}_{gg,q}(y,\mu) \underset{12}{\otimes} \sum_q \big[f_q(\mu) + f_{\bar{q}}(\mu)\big]\,, \qquad (176)$$

which quantifies the contribution of the splitting processes $q \to ggq$ and $\bar{q} \to gg\bar{q}$ to $F^{\text{NLO}}$. Comparing $R^{(q)/(1)}$ with $R^{(2)/(1)}$, we see that the contribution from $V^{(2)}_{gg,q}$ is small compared with the one from $V^{(2)}_{gg,g}$, except when $x_2$ approaches its upper boundary at fixed $x_1$. Interestingly, both for $x_1 = 0.1$ and $x_1 = 0.001$, there is a value of $x_2$ just below $1 - x_1$ at which the contributions from $V^{(2)}_{gg,q}$ and $V^{(2)}_{gg,g}$ cancel each other, whilst each of them separately is of order 10%.

## 5.5 Comparison to other results in the literature

Various papers in the literature contain results for QCD splitting functions at $\mathcal{O}(\alpha_s^2)$ that are (at least) differential in the momentum fractions of two of the produced partons. However, these functions have been derived in somewhat different contexts for other observables. One might wonder whether these objects and the evolution kernels $P_s^{(1)}$ we have computed here are the same. In general, one should not be surprised if there is a difference at $\mathcal{O}(\alpha_s^2)$ even if the functions agree at $\mathcal{O}(\alpha_s)$ — recall that in the single inclusive case, the DGLAP splitting functions appropriate to PDFs (spacelike case) and fragmentation functions (timelike case) differ at $\mathcal{O}(\alpha_s^2)$ [48,49]. It is possible to connect the two beyond LO [48–50, 104–111], but such connections are nontrivial and go beyond simple equality.

In [112, 113], the $\mathcal{O}(\alpha_s^2)$ corrections to $1 \to 2$ splitting functions relevant to the evolution of so-called fracture functions were computed for most partonic channels (excluding, in

---

[7]We use the sets CT14llo and CT14nlo from LHAPDF [101] for $a_s$ and PDFs at LO and NLO, respectively.

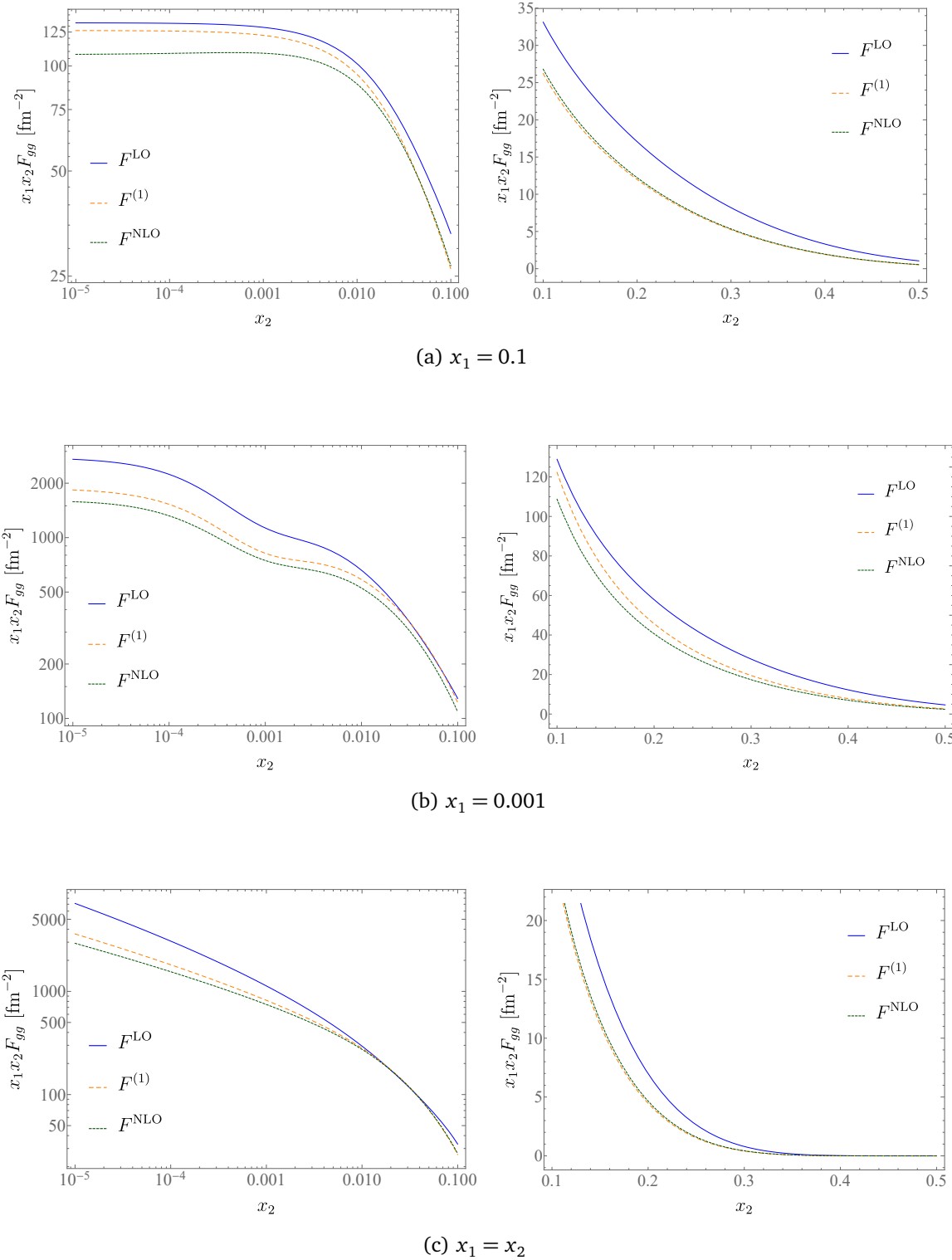

(a) $x_1 = 0.1$

(b) $x_1 = 0.001$

(c) $x_1 = x_2$

Figure 7: Splitting contribution to the two-gluon DPD at $y = 0.022\,\mathrm{fm}$ computed at LO or at NLO, as well as the $\mathcal{O}(a_s)$ part $F^{(1)}$ of the NLO result. The difference between $F^{\mathrm{LO}}$ and $F^{(1)}$ is the order at which $a_s$ and the PDFs at the r.h.s. of (172) are evaluated.

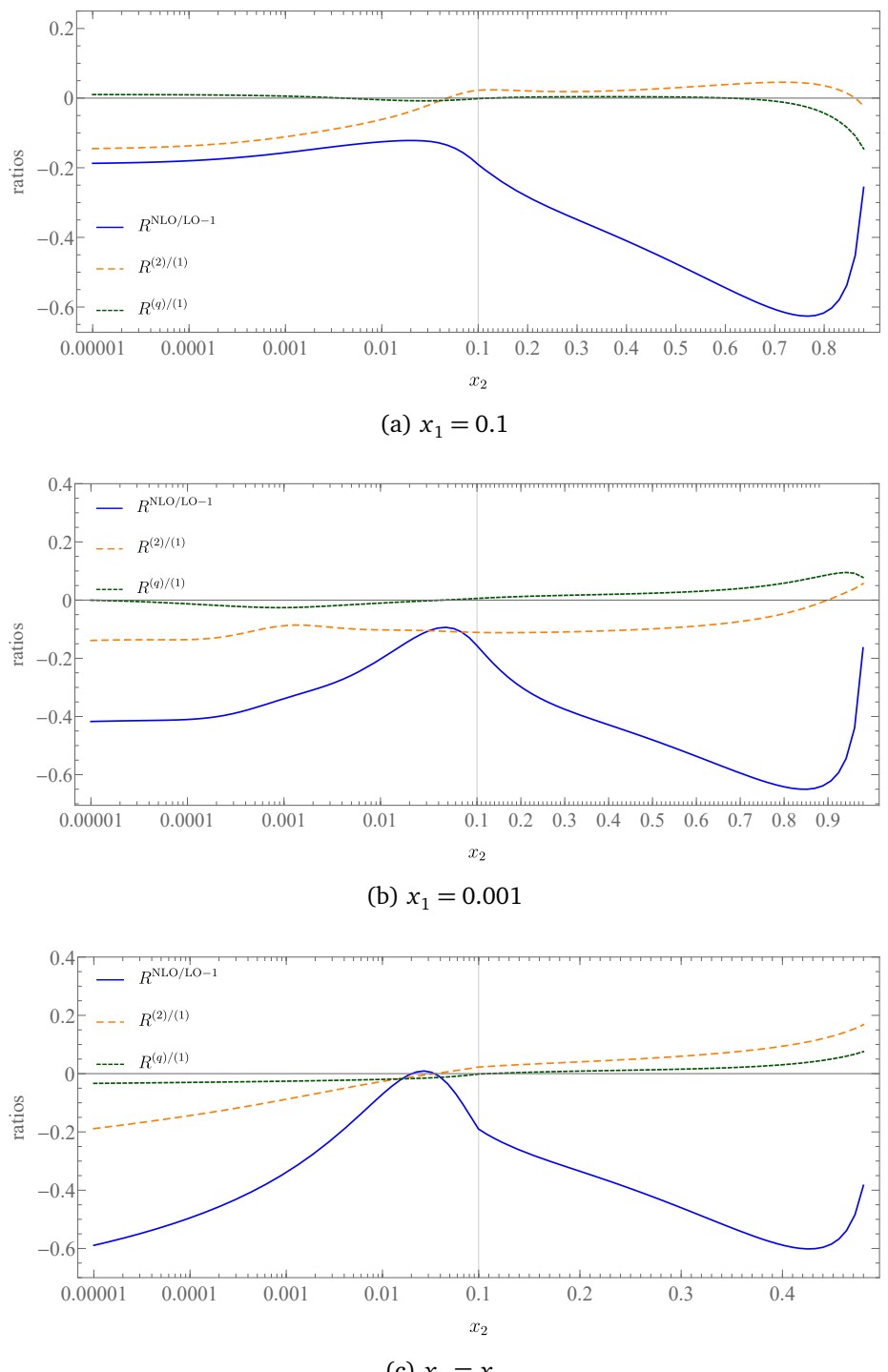

Figure 8: The ratios (174) corresponding to the curves shown in figure 7. Also shown is the ratio $R^{(q)/(1)}$ defined in (176), which quantifies the importance of quark and antiquark splitting contributions to $F^{\mathrm{NLO}}$.

particular, $g \rightarrow gg$). These fracture functions appear in semi-inclusive DIS when we measure the energy of a hadron in the forward region. They have two momentum fraction arguments, one for the parton entering the hard scattering process, and the other for the measured forward hadron. The associated $1 \rightarrow 2$ splitting functions thus involve the measurement of the momentum fraction of one spacelike parton associated with a PDF operator, and that of a timelike parton that ultimately yields the final-state hadron. By contrast, for our DPD splitting functions, we measure the momentum fractions of two spacelike partons.

In [114] it was proposed that the NLO $1 \rightarrow 2$ splitting functions in the evolution of fracture functions are actually equal to the evolution kernels $P_s^{(1)}$ for DPDs, after a simple transformation of the momentum fraction arguments to take into account the fact that one of the momentum fraction arguments in the splitting functions of [112, 113] is not the light-cone momentum of the final parton over that of the initial parton, but that of the final parton over that of the other final parton. We find that this equality does *not* hold, and that in general the transformed splitting functions of [112, 113] do not have the symmetry that our kernels $P_s^{(1)}$ have (and must have) under the interchange of partons 1 and 2. For example, in our results $P_{q\bar{q},g}^{(1)}$ is symmetric under $x_1 \leftrightarrow x_2$, which is due to the equivalent status of partons $a_1$ and $a_2$ together with charge conjugation invariance. By contrast, the same is not true for the corresponding function in [112, 113].

References [115] and [116] give $1 \rightarrow 2$ splitting functions at $\mathcal{O}(\alpha_s^2)$ for the decay of a timelike parton, where one measures the momentum fractions of two of the partons produced. These functions are referred to as "two-body inclusive decay probabilities" in [115, 116]. They appear in the evolution of two-particle fragmentation functions, which in turn are relevant when one measures the energies of two hadrons in an individual jet [117]. We find that in general these functions also differ from the DPD kernels $P_s^{(1)}$. Exceptions to this are the valence-type kernels $P_{q\bar{q},q}^{(1)\,v}$ and $P_{qq,q}^{(1)\,v}$, both of which agree between the spacelike and timelike cases. This is likely due to the fact that for each of these kernels there is only one graph in light-cone gauge, see figure 2m and 2o, which does not have any infrared or ultraviolet subdivergences. This finding is reminiscent of the equality of the single inclusive splitting function $P_{\bar{q}q}^{V\,(1)}$ between the spacelike and timelike cases [48].

Finally, we comment on the tree-level "triple collinear" splitting functions computed in [118, 119]. These quantities are completely unintegrated, depending on the total invariant mass $s_{123}$, the invariant masses of parton pairs $s_{ij}$, as well as the momentum fractions of all partons. They are thus analogous to the initial real emission graphs we compute using FORM or FeynCalc. However, in our calculation we require a momentum shift between amplitude and conjugate amplitude for the two observed partons, whereas the functions computed in [118, 119] have no such shift. Thus we could not have used these functions as the starting point of our calculation for the real emission diagrams. Similar statements apply regarding our virtual diagrams and the one-loop double collinear splitting functions [120–126], which describe the exclusive $1 \rightarrow 2$ splitting process at the one-virtual-loop level. It should be noted that [120–126] also give results for one-loop double collinear "splitting amplitudes", which describe the one-loop splitting process on one side of the cut. We could have potentially used these amplitudes in our calculation. It is worth mentioning that the tree-level triple collinear splitting functions and one-loop double collinear splitting functions can be used as the starting point in many NNLO beam and jet function computations, which do not require a momentum shift between amplitude and conjugate, as was pointed out in [127].

## 5.6 The $\mathcal{N} = 1$ SUSY Relation

There is an interesting relation between the gluon- and quark-initiated single-inclusive splitting functions at $\mathcal{O}(\alpha_s)$ if we set $C_F = C_A = 2T_F = 1$ and $n_F = 1$. Namely, if we define

$$\Delta(x) = \sum_{i=g,q,\bar{q}} [P_{i,g}(x) - P_{i,q}(x)], \tag{177}$$

then we have $\Delta^{(0)}(x) = 0$ [128]. At NLO we have $\Delta^{(1)}(x) \neq 0$ [49], but the expression for $\Delta^{(1)}(x)$ is considerably shorter than those for the individual splitting functions.

This relation holds at LO because when one modifies the colour factors as specified, the theory becomes equivalent to an $\mathcal{N} = 1$ supersymmetric theory, with the quark playing the role of the gluino [129]. We have $\Delta(x) \neq 0$ beyond LO because the conventional methods of dimensional regularisation and $\overline{\text{MS}}$ renormalisation (which are used in [49]) do not preserve supersymmetry. Since the relation is only violated by the regulator, the expressions for $\Delta(x)$ at higher orders are still comparatively small. If one instead uses a regularisation method that preserves supersymmetry, such as dimensional reduction [130], then $\Delta(x) = 0$ holds beyond LO, as was shown for the NLO case in [108, 131–134].

It is interesting to see if a corresponding relation is "approximately" observed by our $1 \to 2$ evolution kernels, where we recall that these have also been obtained using dimensional regularisation and $\overline{\text{MS}}$ renormalisation. The straightforward generalisation of the difference (177) to this case is

$$\Delta(x_1, x_2) = \sum_{i,j=g,q,\bar{q}} [P_{ij,g}(x_1, x_2) - P_{ij,q}(x_1, x_2)]. \tag{178}$$

The relation between $P_s^{(0)}$ and $P^{(0)}$ readily implies that $\Delta^{(0)}(x_1, x_2) = 0$. With the NLO kernels given earlier in this section, we find

$$\Delta^{(1)}(x_1, x_2) = \left\{ \left[ -\frac{p_{qg}(x_1)}{[x_3]_+} - \frac{1 - 5x_1 + 4x_1^2 - 6x_2^2}{2\bar{x}_1^2} \right] + \{\text{all permutations of } x_1, x_2, x_3\} \right\}$$
$$+ \frac{6\log\bar{x}_1 + 6\log x_1 - 7}{3} p_{qg}(x_1)\,\delta(x_3), \tag{179}$$

where it is understood that the plus prescription in the first term is dropped when $x_3$ is replaced by $x_1$ or $x_2$. This is indeed a much simpler expression than those for the individual evolution kernels. Note in particular that for $x_3 \neq 0$, the logarithmic terms completely cancel in $\Delta^{(1)}(x_1, x_2)$, and only the single logarithmic terms survive at $x_3 = 0$. Presumably we would obtain $\Delta^{(1)}(x_1, x_2) = 0$ if we were to convert our results to the dimensional reduction scheme, but to verify this goes beyond the scope of the present study.

# 6 Conclusion

We computed at two-loop order the perturbative matching kernels $V_{a_1 a_2, a_0}(x_1, x_2, y)$ between position space DPDs $F_{a_1 a_2}(x_1, x_2, y)$ and PDFs $f_{a_0}(x)$, as well as the matching kernels $W_{a_1 a_2, a_0}(x_1, x_2, \Delta)$ between momentum space DPDs $F_{a_1 a_2}(x_1, x_2, \Delta)$ and PDFs $f_{a_0}(x)$. The computation of $W_{a_1 a_2, a_0}$ is more involved than the one of $V_{a_1 a_2, a_0}$ because it requires results for bare graphs at one higher order in $\epsilon$. We showed that at two-loop accuracy one can readily extract from $W_{a_1 a_2, a_0}$ the kernels $V_{a_1 a_2, a_0}$, the $1 \to 2$ evolution kernels $P_{a_1 a_2, a_0}(x_1, x_2)$ in the inhomogeneous term of the evolution equation for $F_{a_1 a_2}(x_1, x_2, \Delta)$, as well as the matching coefficients for computing $F_{a_1 a_2}(x_1, x_2, \Delta)$ from $F_{a_1 a_2}(x_1, x_2, y)$. We obtained results for all

possible partonic channels, i.e. all possible combinations of $a_0, a_1$, and $a_2$, while limiting ourselves to unpolarised, colour singlet DPDs. These quantities are needed to compute DPS cross sections at next-to-leading order.

We performed the calculation using both Feynman and light-cone gauge, finding agreement between the results in the two gauges. Loop integrals were performed using the method of differential equations, together with boundary conditions computed using the method of regions and standard integration using Feynman parameters. Each master integral was checked numerically at 10 values of $(x_1, x_2)$.

We extracted the behaviour of the matching coefficients in the threshold limit $x_1 + x_2 \to 1$ and in different limits with small momentum fractions, namely for $x_1 + x_2 \ll 1$, for $x_1 \ll 1$, and for $x_1 \ll x_1 + x_2 \ll 1$. We verified that our two-loop results for $P_{a_1 a_2, a_0}(x_1, x_2)$ obey the number and momentum sum rules given in [54]; this can be viewed as a cross check of our results, or as an explicit verification of these sum rules at the two-loop level. In [114] a relation was proposed between the kernels $P_{a_1 a_2, a_0}$ and the $1 \to 2$ splitting functions appearing in the evolution of fracture functions. We found that this relation is not fulfilled at two-loop order.

It is quite straightforward to extend the results of the present paper to polarised DPDs. Similar methods as shown here can be used to compute the matching coefficients for colour interference. In this case, one needs to explicitly deal with rapidity divergences and soft factors, as briefly described at the end of section 4.2.1. Work in this direction is underway. On the phenomenological side, the numerical example in section 5.4 shows that the difference between two-gluon DPDs computed at leading and at next-to-leading order can be appreciable, in particular at small momentum fractions and for $x_1 + x_2$ close to 1. It will be interesting to investigate in a more comprehensive way how the two-loop corrections computed in this paper affect DPDs, double parton luminosities, and ultimately the cross sections of processes that are sensitive to DPS.

# Acknowledgements

We gratefully acknowledge discussions with Johannes Michel. Two of us (MD and JRG) thank the Erwin Schrödinger International Institute for Mathematics and Physics (ESI) for hospitality during the Parton Showers, Event Generators and Resummation Workshop (PSR19), when portions of this work were completed. The figures in this work were produced with Jaxo-Draw [135, 136].

We would like to dedicate this work to the memory of James Stirling.

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
