# Peer review of "Two-loop splitting in double parton distributions"

_SciPost Physics, doi:SciPost Phys. 7, 017 (2019)_

## Round 1 · Referee Report · Anonymous (Referee 1) · 2019-4-14

Strengths

The paper given an extremely careful account of these splitting function, including analytic results, details of the calculations, and a comparison with the literature.

Weaknesses

The discussion is a little long in many places. Like for example when the authors introduce dimensional regularization, convolutions, the running strong couplings, ect. Not that introducing these basics is not useful, but the level of care makes the paper a little hard to read. More importantly, the paper is too brief on the conceptual discussion and on the physics impact.

Report

The paper is written in a very specific style, and I cannot say I personally like it. What I am missing is some kind of discussion of the concepts behind the calculation, so readers see where things are going in each chapter. And, more importantly, I am missing a discussion of the impact of these results. If this is a paper about LHC physics, why should for instance an interested experimentalist care? This is completely unclear in this otherwise very interesting and definitely very clear paper.

Requested changes

I would love to see a motivational discussion at the very beginning, accessible to an experimentalist or an incoming PhD student in the field; and another section 5.6 with a discussion of the results and their impact. Why do we have to know those splitting kernels and how would this calculation make for a better agreement between theory and data? The paper itself I find hard to read, but it's the way the authors want to write it, so I am fine with that.

---

## Round 1 · Referee Report · Anonymous (Referee 2) · 2019-4-21

Strengths

A detailed analysis of DPD renormalization and sum rules

Weaknesses

A bit technical, but that is the nature of the subject

Report

The paper discusses the NLO renormalization of DPDs (double parton distribution functions), and the mixing of these with ordinary PDFs (i.e. single parton distribution functions). The paper makes important contributions, and should be published. The paper is rather technical, and only suitable for those studying DPDs in detail.

Requested changes

None

---

## Round 1 · Referee Report · Anonymous (Referee 3) · 2019-5-22

Strengths

1- Very well written 2- Good introduction and motivation 3- Level of detail in the explanation of calculations 4- Overall structure meaningful 5- Can follow the ideas throughout despite the paper being very long 6- Very relevant calculation for this field 7- Detailed discussion of results overall and in relation to earlier work 8- Balance of technical detail vs explanatory text well chosen

Weaknesses

If any weakness is to be pointed out, then perhaps hints towards next steps where the results are to be applied in relation to possible measurements and accuracies at the upcoming high luminosity phase of the LHC.

Report

An important computation is being reported on in this paper which documents a step forward in the field of double Parton scattering. The computation is very timely and is in line with the expected performance of the upcoming LHC high luminosity run. Despite being technical, this paper will nevertheless be an interesting read for students and postdocs in this and related fields as many details are explained very well.

Requested changes

No changes requested.

---

## Round 2 · Author Response

We thank the referees for their careful reading of our manuscript and for the constructive remarks. Let us reply to the reports of Referees 1 and 3 in turn.

Referee 1

System Message: WARNING/2 (<string>, line 4)

Title underline too short.

Referee 1
* * *
Indeed, the paper gives a fair amount of technical detail and is not short. However, we believe that this is justified: we give as much detail as we think is necessary for a QCD practitioner to understand what we compute and how we do it, in the spirit of our work being reproducible. The formulae regarding dimensional regularisation and the running coupling are given to set up our notation and to make the paper self consistent (rather than forcing a reader to look elsewhere for definitions that are needed to read the present manuscript). The same holds for convolutions, which are in a large part non-standard because they involve functions of two momentum fractions.

Concerning a lack of conceptual discussion mentioned by the referee, we think that we present the relevant methods as they are used for our specific calculation. A more general discussion, e.g. of the theory of double parton scattering would go beyond the scope of our work and can be found elsewhere in the literature. We have augmented the references given in the introduction, such that an interested reader can more easily locate the relevant papers.

The introductory paragraph we added at the beginning of section 2 is meant to guide the reader by giving an overview of what is to come in that section.

As to the physics impact of our calculation, we have added a section 5.4, where we present a numerical example for NLO effects in double parton distributions. An analysis at the level of physical cross sections will involve a significant number of further steps. Performing and presenting such calculations is a project in itself, much beyond the scope of the present work.

To provide a better motivation for our work, we have extended the introduction, giving more specific reasons to study DPS and to analyse it at NLO accuracy. For a more general or more extended discussion of DPS, our paper would not be the most suitable place. We instead point the reader to the existing literature, especially to the monograph [39], which also covers phenomenological and experimental aspects.

We would like to point out that, by the nature of the presented material, the present work will be most interesting and accessible to theorists working on DPS - and possibly to practitioners of higher-order computations - rather than to experimentalists. In this respect, we agree with the assessment in the Report of Referee 2.

Referee 3

System Message: WARNING/2 (<string>, line 19)

Title underline too short.

Referee 3
* * *
In the last paragraph of the Conclusion, we now give a bit more detail about where we stand and what are the next steps for studying parton splitting in DPS. Some particularly interesting DPS processes are identified in the introduction.

We hope that our revised manuscript addresses the referee reports in an adequate manner and send our best regards,

The authors

---

## Round 2 · List of Changes

1. In the introduction (p.3) we have added some general physics motivation to study double parton scattering (DPS), along with references to experimental results (thus providing a connection to phenomenology). For a broader introduction to DPS, we refer to a recent monograph (Ref [39]).

2. Also in the introduction (p.4) we give more arguments for NLO computations of DPS in general, and the perturbative splitting mechanism in particular.

3. For the general orientation of the reader, we have added a paragraph at the beginning of section 2 (p.5), and a sentence at the end of the first paragraph of section 4 (p.21).

4. In a new section 5.4, we give a numerical illustration of the difference between a double parton distribution computed at leading and at next-to-leading order. While this is for a particular parton combination in particular kinematics, it shows that the difference between the two perturbative orders can be numerically important. We are aware that this is not equivalent to a study of DPS cross sections at NLO, mentioned by Referee 1. Such a study will require significant additional work - well beyond the computation of the NLO kernels - and we think it is justified to leave this to a future project.

5. In the last paragraph of the Conclusion, we give a slightly more detailed outlook on future work.

---

## Editorial Decision

published